# Post-Training Augmentation Invariance

**Keenan Eikenberry**                                        *Keenan.J.Eikenberry@dartmouth.edu*
*Department of Mathematics, Dartmouth College*

**Lizuo Liu**                                                      *Lizuo.Liu@dartmouth.edu*
*Department of Mathematics, Dartmouth College*

**Yoonsang Lee**                                              *Yoonsang.Lee@dartmouth.edu*
*Department of Mathematics, Dartmouth College*

**Reviewed on OpenReview:** *https://openreview.net/forum?id=Z4uUwU6zRe*

## Abstract

This work develops a framework for post-training augmentation invariance, in which our goal is to add invariance properties to a pretrained network without altering its behavior on the original, non-augmented input distribution. We define this notion precisely and additionally introduce augmented encoders, which are probabilistic encoders that formalize augmentation-based encoding processes and that serve as our fundamental object of study. We introduce two losses for augmented encoders, namely, Markov-Wasserstein minimization and Wasserstein correlation maximization, and we demonstrate empirically that both losses can be used to train lightweight, one-hidden-layer MLP adapter networks $E_\theta$ that, when appended to the latent space of a pretrained network $F$, do indeed lead to (approximate) post-training augmentation invariance. For example, on STL10 with $F$ = DINO features, the composite network $C \circ E_\theta \circ F$, where $C$ is a linear classifier and where $E_\theta$ is one of our proposed adapter networks, achieves 94% classification accuracy on arbitrarily rotated images, whereas a network of the form $C \circ F$ without the adapter $E_\theta$ drops to 71% accuracy. Similarly, we can boost noise-invariant classification results from 58% up to 86%. Significantly, we obtain these results with no fine-tuning (the weights of $F$ remain frozen throughout), and our methods introduce little corruption to the original features, since $E_\theta$ acts nearly isometrically on the non-augmented latent distribution. In contrast, we show that adapter networks trained with alternative candidate losses, specifically SimCLR and HSIC maximization, produce uncompetitive classification results and fundamentally corrupt the original latent space. Code available at `https://github.com/keenan-eikenberry/augmentation_invariance`.

## 1 Introduction

Large-scale pretrained models have become the foundation of modern machine learning pipelines. For vision, models such as vision transformers (Dosovitskiy et al., 2020), CLIP (Radford et al., 2021), DINO (Caron et al., 2021), and masked autoencoders (He et al., 2022) are trained on massive datasets and provide general-purpose features that tend to transfer well to a variety of downstream tasks, including classification, object detection, feature matching, segmentation, and more. However, pretrained models may not be invariant to augmentations that could be critical for various applications. For example, motion estimation or feature matching for satellite or medical images may require features that are invariant to rotations, or perhaps to affine or perspective transformations, more generally. We may also need strong robustness to noise or adversarial perturbations, both of which may be regarded as non-invertible augmentations that introduce information loss. However, any given model is not necessarily guaranteed to satisfy these requirements.

The motivating question of this work, then, is as follows: **Can we make a pretrained model invariant to specific augmentations post-training without corrupting its existing capabilities?** This problem,

which we call **post-training augmentation invariance**, is distinct from general representation learning and poses unique challenges. Retraining foundation-scale models to be invariant to various augmentations is computationally prohibitive and risks degrading the learned representations, which may depend on particular augmentation pipelines for success, such as the central use of crops in many self-supervised learning frameworks. Additionally, Kumar et al. (2022) show that standard fine-tuning methods can distort pretrained features, sometimes significantly. They develop an effective method for combining linear probe methods with full fine-tuning, but we will show in this work that, at least for the goal of augmentation invariance, even simpler methods can dramatically improve performance without any fine-tuning at all. That is, we leave the weights of the pretrained network completely frozen and unchanged and instead only append lightweight adapter networks to the latent space.

## 1.1 Contributions

In this work, we provide a comprehensive framework and initial solution to the problem of post-training augmentation invariance. Though our formal definitions are math-heavy and require machinery from measure theory and optimal transport, we highlight the following experimental results to emphasize the practical value of our work: On STL10 with $F = \text{DINO}$ features, the composite network $C \circ E_\theta \circ F$, where $C$ is a linear classifier and where $E_\theta$ is one of our proposed, lightweight adapter networks, achieves 94% classification accuracy on arbitrarily rotated images, whereas a network of the form $C \circ F$ without the adapter $E_\theta$ drops to 71% accuracy. Similarly, we can boost noise-invariant classification results from 58% up to 86%. We obtain these results with no fine-tuning (the weights of $F$ remain frozen throughout), and, perhaps most importantly, we achieve these significant increases in accuracy without corrupting the original feature space: $E_\theta$ acts nearly isometrically on the non-augmented latent distribution.

Our exact contributions are as follows:

1. We formalize the problem of post-training augmentation invariance through the notion of $(t, \mu_X, F, V)$-invariance (Definition 3.4), which requires that an adapter $E_\theta$ makes the composite network $E_\theta \circ F$ invariant to an augmentation $t$ while preserving the structure of the pretrained latent distribution $F_\sharp \mu_X$ up to a map $V$ in some admissible class of maps $\mathcal{V}$.

2. We introduce augmented encoders (Section 3.1), which are probabilistic encoders that formalize the process of randomly augmenting and then encoding an input, as a fundamental object of study and as a general framework for augmentation-based learning, independent of any specific training objective.

3. We propose two training objectives for augmented encoders that can be used for post-training augmentation invariance, namely, an anchored mean-squared error (MSE) loss, which can also be viewed as a minimization problem in a generalized $Lp$ metric for Markov-Wasserstein kernels (Section 3.3.1), and Wasserstein correlation maximization (Section 3.3.2), which maximizes the Wasserstein correlation of the joint distribution induced by the augmented encoder.

4. We demonstrate empirically (Section 4) for various pretrained networks $F$ that simple, one-hidden-layer multilayer perceptrons (MLPs) $E_\theta$ trained on our proposed losses make the composite network $E_\theta \circ F$ approximately $(t, \mu_X, F, V)$-invariant for an approximate, local isometry $V$. By contrast, we show that two other candidate losses, namely, SimCLR and a Hilbert-Schmidt Independence Criterion (HSIC) maximization loss, produce uncompetitive invariance results and significantly corrupt the pretrained latent space.

## 1.2 Related Work

### 1.2.1 (Self-Supervised) Representation Learning

Many self-supervised learning objectives, such as SimCLR (Chen et al., 2020), MoCo (He et al., 2020), SwAV (Caron et al., 2020), BYOL (Grill et al., 2020), and DINO (Caron et al., 2021), learn representations by maximizing agreement between differently augmented views of the same input. These approaches naturally

induce (approximate) invariance to their chosen augmentations and have achieved remarkable success in pretraining. However, as we demonstrate empirically, applying a SimCLR loss (which we take to be a suitable stand-in for contrastive losses, more generally) in a post-training setting fundamentally alters the geometry of the original latent space, making the pretrained features subsequently unusable. This is unsurprising: Contrastive methods cluster semantically similar examples, which is ideal for learning representations from scratch, but inappropriate when the goal is to preserve existing structure.

Other work in representation learning more directly employs variants of the InfoMax principle first introduced by Linsker (1988) to maximize mutual information (MI) between input and latent distributions; see, for example, Bachman et al. (2019); Hjelm et al. (2018); Hu et al. (2017); Oord et al. (2018); Rezaabad & Vishwanath (2020); Zhao et al. (2019), and, in particular, see Tschannen et al. (2019) for a unified perspective on many of these works. MI estimation, of course, is computationally difficult in high-dimensions, and, in practice, neural estimators, as defined in Belghazi et al. (2018) or Poole et al. (2019), for example, are used to estimate lower bounds on the quantity. However, as shown in McAllester & Stratos (2020), high-confidence, distribution-free lower bounds of any type have exponential sample complexity in the size of the bound. More strikingly, Tschannen et al. (2019) show that maximizing MI does not necessarily lead to effective representations for various downstream tasks and that the success of MI-based objectives involves a subtle interplay between encoder architectures and MI estimators.

One of our initial motivations for undertaking this work was to explore the potential suitability of Wasserstein dependence, or its normalized version, Wasserstein correlation, as a general drop-in replacement for mutual information in MI-based representation learning methods. While Ozair et al. (2019) showed that Wasserstein dependence maximization can be used for general representation learning when combined with additional regularization terms, we found in our own investigations that Wasserstein dependence alone, or, more specifically, Wasserstein correlation maximization, cannot, in fact, be used as generic replacement for MI-maximization, as explained further in Remark 3.5. Briefly, Wasserstein correlation maximization inverts the dichotomy introduced before: It (approximately) preserves the metric structure of the input distribution, which is therefore fatal when the goal is to cluster data according to semantic similarity, but ideal when trying to leave a pretrained latent space intact.

To test whether other measures of statistical dependence might perform similarly to our Wasserstein correlation maximization objective, we also consider maximizing the Hilbert-Schmidt Independence Criterion (HSIC) (Gretton et al., 2005), which has been shown to be effective for self-supervised learning (Li et al., 2021). As will be shown experimentally, though, the HSIC loss, like the SimCLR loss, is also uncompetitive and corrupts the pretrained latent space even more dramatically. This suggests that the geometric properties of the optimal transport-based objectives are playing a non-trivial role in achieving post-training augmentation invariance.

### 1.2.2 Fine-Tuning and Model Adaptation

The overall aim of our work is related to fine-tuning (Kumar et al., 2022; Goyal et al., 2023; Wortsman et al., 2022; Xin et al., 2024), but our methods leave the pretrained network frozen and unaltered. We seek instead to develop adapter networks, which have been explored for various use-cases, such as improved semantic segmentation or domain generalization (Dukler et al., 2023; Ji et al., 2025; Kim et al., 2021; Rebuffi et al., 2017; Xu et al., 2023). To our knowledge, no one has developed a general framework comparable to ours for adapter networks that can achieve (approximate) post-training augmentation invariance for arbitrary augmentation types.

### 1.2.3 Optimal Transport-Based Methods

Our work is related, more broadly, to the many uses of optimal transport in machine learning. Transport-based methods have been influential in a number of areas, including generative modeling (Arjovsky et al., 2017; Bousquet et al., 2017; Kolouri et al., 2019; Kunkel & Trabs, 2024; Liutkus et al., 2019; Nadjahi et al., 2019; Nguyen et al., 2022; Tolstikhin et al., 2017; Wu et al., 2019), domain adaptation (Courty et al., 2016; 2017; Lee et al., 2019; Shen et al., 2018), and adversarial robustness (Bhagoji et al., 2019; Bai et al., 2023; Levine & Feizi, 2020; Nguyen et al., 2023; Wong et al., 2019; Wu et al., 2020). Our use

of Wasserstein correlation, in particular, builds on a growing body of work on transport-based statistical dependency measures (Liu et al., 2022; Móri & Székely, 2020; Mordant & Segers, 2022; Nies et al., 2021; Wiesel, 2022). These ideas have been applied to feature selection (Li et al., 2023) and representation learning (Ozair et al., 2019; Xiao & Wang, 2019), but to our knowledge, no one has used Wasserstein correlation, nor our other transport-based loss, for invariance learning, particularly in a post-training setting where additional structure-preservation constraints are needed.

### 1.3 Paper Organization

The rest of the paper is organized as follows: In Section 2, we present background material needed for the rest of the paper. The material on optimal transport is standard, but the material on Markov-Wasserstein kernels (Section 2.1) and Wasserstein correlation (Section 2.2) is likely less well-known. Our main contributions are presented in Section 3, and our experimental results are presented in Section 4. Finally, limitations of the current framework and suggestions for future work are given in Section 5.

## 2 Background

We recall all standard definitions from optimal transport (Wasserstein distances, couplings, product measures, sliced Wasserstein distances) in Appendix A. Here, we introduce only the two constructions that are essential for defining our loss functions: the generalized $L^p$ metric for Markov-Wasserstein kernels (Section 2.1) and Wasserstein correlation (Section 2.2). Readers interested primarily in the practical aspects of this work can skip directly to Section 3.3.1 for a self-contained presentation of our main loss function without reference to optimal transport machinery.

### 2.1 Markov-Wasserstein Kernels

We model encoders abstractly as Markov-Wasserstein kernels, which are Markov kernels valued in Wasserstein spaces (see Appendix A for the full development, including Markov kernels, disintegrations, and Bayesian inverses). For our purposes, the key construction is the generalized $L^p$ metric between Markov-Wasserstein kernels, which gives rise to our Markov-Wasserstein minimization loss.

**Definition 2.1** *Let $(X, d_X, \mu_X)$ be a metric measure space (i.e., a Polish metric space equipped with a probability measure $\mu_X$ on the Borel $\sigma$-algebra induced by $d_X$), and let $(Y, d_Y)$ be a Polish metric space. The* **$L^p$ space** *$L^p_{\mu_X}(X, Y)$ is the set of $\mu_X$-a.e. equal equivalence classes of measurable functions $f : X \to Y$ satisfying*

$$\int_X d_Y^p\big(y_0, f(x)\big)\mu_X(dx) < \infty \tag{1}$$

*for some (and hence any) $y_0 \in Y$. We say that $f$ is* **$L^p$** *integrable if this condition is met. Further, $L^p_{\mu_X}(X, Y)$ is a metric space when equipped with the* **$L^p$** *metric*

$$d_{L^p} : L^p_{\mu_X}(X, Y) \times L^p_{\mu_X}(X, Y) \to \mathbb{R}_{\geq 0} \tag{2}$$

*defined by*

$$d_{L^p}(f, g) = \left(\int_X d_Y^p\big(f(x), g(x)\big)\mu_X(dx)\right)^{1/p} \tag{3}$$

*for $1 \leq p < \infty$.*

When $Y$ is itself a Wasserstein space $\mathcal{P}_p(Y)$ and $d_Y$ is the Wasserstein metric $W_{p,d_Y}$, we call elements of $L^p_{\mu_X}(X, \mathcal{P}_p(Y))$ **Markov-Wasserstein kernels**. Also, we denote the corresponding $L^p$ metric by $\mathrm{MW}_p$ and call it the **Markov-Wasserstein metric**. Once again, see Appendix A for further details.

### 2.2 Wasserstein Correlation

Here we introduce Wasserstein correlation, which we use to define the Wasserstein correlation maximization loss in Section 3.3.2. First, recall that the mutual information of random variables $X$ and $Y$ is defined

as $I(X;Y) = D_{KL}(P_{XY} \| P_X \otimes P_Y)$, where $D_{KL}$ is KL-divergence, and where $P_{XY}$ and $P_X \otimes P_Y$ are, respectively, the joint and product distributions of the pair $(X, Y)$. Wasserstein dependence is the natural optimal transport-based analog of this quantity.

**Definition 2.2** *Let $(X, d_X)$ and $(Y, d_Y)$ be Polish metric spaces (or simply Euclidean spaces in practice), and let $(X \times Y, d_{XY})$ be the standard product space equipped with the product metric of order p for some $1 \leq p < \infty$. Then, the **Wasserstein dependence of order p** on $\mathcal{P}_p(X \times Y)$*

$$WD_p : \mathcal{P}_p(X \times Y) \to \mathbb{R}_{\geq 0} \tag{4}$$

*is defined by*

$$WD_p(\pi) = W_{p, d_{XY}}(\pi, \pi^1 \otimes \pi^2), \tag{5}$$

*where $\pi^1 \in \mathcal{P}_p(X)$ and $\pi^2 \in \mathcal{P}_p(Y)$ are the first and second marginals of $\pi$.*

We cannot train models directly on Wasserstein dependence maximization, since the encoder can trivially hack the objective by spreading the latent codes arbitrarily far apart. One possibility is to add a prior-matching term in the latent space. This is the approach taken by information-maximizing variational autoencoders (Rezaabad & Vishwanath, 2020; Zhao et al., 2019). In these and similar models, the support of the latent distribution remains bounded by being constrained to remain close, in some divergence or metric, to a prior distribution, typically a Gaussian. In our case, we work instead with a normalized version of dependence, namely, Wasserstein correlation, which constrains the variance of the latent codes.

**Definition 2.3** *Take the same setup as in Definition 2.2, and let $1 \leq p < \infty$ be an integer. Then, the **Wasserstein correlation of order p** on $\mathcal{P}_p(X \times Y)$*

$$WC_p : \mathcal{P}_p(X \times Y) \to \mathbb{R}_{\geq 0} \tag{6}$$

*is defined by*

$$WC_p(\pi) = \frac{WD_p(\pi)}{\left(WD_p(\pi_D^1) \cdot WD_p(\pi_D^2)\right)^{1/p}}, \tag{7}$$

*where $\pi_D^1$ is the diagonal distribution on $\mathcal{P}_p(X \times X)$ given by $\pi_D^1(A \times B) = \pi^1(A \cap B)$ for $A \times B \in \Sigma_{X \times X}$, and similarly for $\pi_D^2 \in \mathcal{P}_p(Y \times Y)$.*

In the above, the terms in the denominator are self-variance terms that measure how spread out the marginal distributions are, and the dependence scores are computed in the Wasserstein spaces $(\mathcal{P}_p(X \times X), W_{p, d_{XX}})$ and $(\mathcal{P}_p(Y \times Y), W_{p, d_{YY}})$, respectively. We leave this implicit to reduce notational clutter. These normalization terms guarantee that Wasserstein correlation is bounded between zero and one.

For computational implementations, we simply substitute the sliced Wasserstein distance for the ordinary Wasserstein distance in the definitions above. Naturally, we call the resulting quantities **sliced Wasserstein dependence** and **sliced Wasserstein correlation**, and we use $\mathrm{SD}_p$ and $\mathrm{SC}_p$ for the corresponding notation. In practice, we work (in the non-augmented case) with empirical joint distributions corresponding to pairs $(x_i, E_\theta(x_i))$, and we approximate product distributions by shuffling coordinates, as in Li et al. (2023). Altogether then, we have

$$\mathrm{SC}_p\left(\frac{1}{N} \sum_{i=1}^N \delta_{\left(x_i, E_\theta(x_i)\right)}\right) = \frac{SW_p\left(P^N(X, Z), P_{\sigma_{XZ}}^N(X, Z)\right)}{\left(SW_p\left(P^N(X, X), P_{\sigma_{XX}}^N(X, X)\right) \cdot SW_p\left(P^N(Z, Z), P_{\sigma_{ZZ}}^N(Z, Z)\right)\right)^{1/p}}, \tag{8}$$

where

$$P^N(X, Z) = \frac{1}{N} \sum_{i=1}^N \delta_{\left(x_i, E_\theta(x_i)\right)} \quad \text{and} \quad P_{\sigma_{XZ}}^N(X, Z) = \frac{1}{N} \sum_{i=1}^N \delta_{\left(x_{\sigma_1(i)}, E_\theta(x_{\sigma_2(i)})\right)} \tag{9}$$

for random permutations $\sigma_k$ on $\{1, \ldots, N\}$ and for $Z = E_\theta(X)$. The other quantities are defined similarly. Also, note that it is sufficient to only shuffle one coordinate to approximate the product distribution, but the estimate is generally better when shuffling both.

## 3 Methods

Anyone interested primarily in the practical aspects of this work can skip to Section 3.3.1 for a straightforward presentation of our main loss function, which we have found to be effective in a broad range of scenarios. In Sections 3.1 and 3.2, we formalize augmentation-based learning processes with Markov-Wasserstein kernels and give precise definitions for post-training augmentation invariance. It is not necessary to follow this material, though, to understand our experimental results or to apply the ideas in practice.

### 3.1 Augmented Encoders

Here we introduce our fundamental object of study, namely augmented encoders, which are probabilistic encoders that formalize the process of randomly augmenting and then encoding an input. Then, in Section 3.3, we define two loss functions for invariance learning with augmented encoders.

Define an **augmentation** to be a (measurable) map of the form $t : A \times X \to X$, where $A$ is the parameter space, typically taken to be (a subset of) some (potentially factored) Euclidean space $\mathbb{R}^{p_1} \times \cdots \times \mathbb{R}^{p_\ell}$. Note that this includes composite augmentations since we can consider, for example, $t\big((a_1, a_2), x\big) = t_2\big(a_2, t_1(a_1, x)\big)$, and likewise for more general compositions $t_k \circ t_{k-1} \circ \cdots \circ t_1$. Now, let $T = \{t_1, \ldots, t_m\}$ be a collection of $m$ augmentations, let $w = (w_1, \ldots, w_m)$ be a collection of positive weights with $\sum_{j=1}^{m} w_j = 1$, let $\nu = (\nu_1, \ldots, \nu_m)$ be a collection of distributions $\nu_j \in \mathcal{P}_p(A_j)$ on the parameter spaces $A_j$, and let $E : X \to Z$ be a deterministic encoder, which we simply take to be a measurable map, though, in practice, it will have more or less smoothness properties. Then, we define the **augmented encoder with respect to** $(T, w, \nu)$, denoted simply by $E^T : X \to \mathcal{P}_p(Z)$, by

$$E^T(x) = \sum_{j=1}^{m} w_j \int_{A_j} \delta_{E(t_j(a,x))} \, \nu_j(da), \tag{10}$$

where, for $x_0 \in X$, the distribution $E^T(x_0) \in \mathcal{P}_p(Z)$ is defined on test functions $\phi \in C_b(Z)$ by

$$\int_Z \phi(z) E^T(x_0)(dz) = \int_Z \phi(z) \left( \sum_{j=1}^{m} w_j \int_{A_j} \delta_{E(t_j(a,x_0))} \nu_j(da) \right) (dz) \tag{11}$$

$$= \sum_{j=1}^{m} w_j \int_{A_j} (\phi \circ E)\big(t_j(a, x_0)\big) \nu_j(da).$$

Empirically, if $\nu_j$ is approximated with $n_j$ samples as $\frac{1}{n_j} \sum_{k=1}^{n_j} \delta_{a_j^k}$, then $E^T(x_0)$ is approximated as

$$E^T(x_0) \approx \sum_{j=1}^{m} \sum_{k=1}^{n_j} \frac{w_j}{n_j} \delta_{E(t_j(a_j^k, x_0))}. \tag{12}$$

Additionally, if $\mu_X$ is approximated empirically as $\frac{1}{n} \sum_{i=1}^{n} \delta_{x_i}$, then the generalized pushforward distribution (see definition A.14), or latent distribution, $E^T \odot \mu_X$ is approximated by

$$E^T \odot \mu_X \approx \sum_{i=1}^{n} \sum_{j=1}^{m} \sum_{k=1}^{n_j} \frac{w_j}{n n_j} \delta_{E(t_j(a_j^k, x_i))}. \tag{13}$$

In practice, given a batch of data $\{x_i\}_{i=1}^{N}$, we only sample $s$ times from (12) for each $x_i$ (typically with $s = 3$), and we also allow the augmentation parameters to vary across the batch, giving us

$$E^T \odot \left( \frac{1}{N} \sum_{i=1}^{N} \delta_{x_i} \right) \approx \frac{1}{sN} \sum_{i=1}^{N} \sum_{k=1}^{s} \delta_{z_i^k}, \tag{14}$$

where $\{z_i^1, \ldots, z_i^s\}$ is a collection of $s$ independent samples from $E^T(x_i)$.

Strictly speaking, we have not yet guaranteed that $E^T$ is measurable under definition (10). We will bypass a full consideration of this issue here, since, in practice, we only work with empirical distributions. Formally, though, this can be done by applying Fubini-Tonelli analogs in the Markov-Wasserstein setting.

### 3.2 Augmentation Invariance

Let $\mu_X \in \mathcal{P}_p(X)$ be a distribution, and let $t : A \times X \to X$ be an augmentation. We want encoders that are invariant to $t$, but we must first define invariance carefully, since literal mathematical invariance may immediately imply collapse of (parts of) the representation space. In particular, if there are **collisions**, namely distinct points $x_1$ and $x_2$ in supp($\mu_X$) with either $t(a_1, x_1) = x_2$ for some augmentation parameter $a_1$, or else $t(a_1, x_1) = t(a_2, x_2)$ for augmentation parameters $a_1$ and $a_2$, then exact invariance would imply

$$E(x_1) = E\big(t(a_1, x_1)\big) = E(x_2) \tag{15}$$

in the first case, or, similarly,

$$E(x_1) = E\big(t(a_1, x_1)\big) = E\big(t(a_2, x_2)\big) = E(x_2) \tag{16}$$

in the second. As such, we refine our desired notion of invariance as follows.

**Definition 3.1** *Let $\mu_X \in \mathcal{P}_p(X)$ and let $t : A \times X \to X$ be an augmentation with parameter distribution $\nu \in \mathcal{P}_p(A)$. Define the **t-augmentation set** $Aug(x, t)$ of $x$ by*

$$Aug(x, t) = \{t(a, x) : a \in supp(\nu)\}. \tag{17}$$

*Further, define the set of **unique augmentations** $UAug_{\mu_X}(x, t)$ of $x$ with respect to $t$ and $\mu_X$ by*

$$UAug_{\mu_X}(x, t) := Aug(x, t) \cap \left( \bigcup_{w \in supp(\mu_X) \backslash \{x\}} Aug(w, t) \right)^c \cap supp(\mu_X)^c. \tag{18}$$

*That is, $UAug_{\mu_X}(x, t)$ is the set of all collision-free augmentations of $x$.*

In practice, approximate, rather than exact, collisions are the main obstacle to adding invariance properties to a pretrained model. In our experimental results, we show that the following notion of **collision rate** is a useful proxy for the tractability of the optimization problem.

**Definition 3.2** *Let $F : X \to Z$ be an encoder, let $t : A \times X \to X$ be an augmentation with parameter distribution $\nu \in \mathcal{P}_p(A)$, and let $\mu_X \in \mathcal{P}_p(X)$ be a distribution equipped with a class function $c : supp(\mu_X) \to \{1, \ldots, m\}$. The **(augmented-to-clean) collision rate** of $F$ with respect to $t$ and $\mu_X$ is*

$$\mathrm{CR}(F, t, \mu_X) := \mathbb{E}_{\substack{x \sim \mu_X \\ a \sim \nu}} \left[ \mathbf{1} \left( \min_{\substack{w \in supp(\mu_X) \\ c(w) \neq c(x)}} \|F(t(a, x)) - F(w)\| < \min_{\substack{w \in supp(\mu_X) \\ c(w) = c(x)}} \|F(t(a, x)) - F(w)\| \right) \right], \tag{19}$$

*where $\mathbf{1}$ is the indicator function for the event in parentheses. That is, $\mathrm{CR}$ measures the fraction of augmented samples whose nearest non-augmented, or clean, neighbor in the latent space belongs to a different class than their own.*

**Definition 3.3** *Again, let $\mu_X \in \mathcal{P}_p(X)$ and let $t : A \times X \to X$ be an augmentation with parameter distribution $\nu \in \mathcal{P}_p(A)$. A (deterministic) encoder $E : X \to Z$ is $(\mathbf{t}, \mu_\mathbf{X})$-**invariant** (or invariant with respect to $t$ and $\mu_X$) if for all $x \in supp(\mu_X)$ and all $y = t(a, x) \in UAug_{\mu_X}(x, t)$, we have*

$$E(y) = E\big(t(a, x)\big) = E(x). \tag{20}$$

*(We can alternatively say that this condition should hold almost everywhere, but we will ignore subtleties like this.)*

This definition specifies that, for the purposes of representation learning, an encoder should only be invariant with respect to collision-free augmentations, as otherwise (partial) collapse of the latent space immediately follows. Since our primary aim is to add additional invariance properties to pretrained models while preserving the structure of the original latent space up to isometry, or at least up to some class of admissible maps, we introduce the following definition.

**Definition 3.4** *Let $\mu_X \in \mathcal{P}_p(X)$ be a distribution, let $t : A \times X \to X$ be an augmentation with parameter distribution $\nu \in \mathcal{P}_p(A)$, let $F : X \to Z$ be a (pretrained) encoder, and let $\mathcal{V}(Z, W)$ be an admissible class of maps (e.g., isometries, bi-Lipschitz maps, etc.). An encoder $E : Z \to W$ is $(\mathbf{t}, \mu_{\mathbf{X}}, \mathbf{F}, \mathbf{V})$-invariant if $E \circ F$ is $(t, \mu_X)$-invariant and if $(E \circ F)(x) = (V \circ F)(x)$ for all $x \in supp(\mu_X)$ where $V : Z \to W$ is in $\mathcal{V}(Z, W)$.*

That is, Definition 3.4 specifies that the composite encoder $E \circ F$ should be augmentation invariant with respect to $t$ and the data distribution $\mu_X$, but, additionally, we require that $E$ preserve the (metric) structure of the latent distribution $F_\sharp \mu_X$ up to $V$. This is the precise definition of what we mean by **post-training augmentation invariance**: adding additional invariants to the latent space of a pretrained network $F$ without corrupting the structure of that space.

We stress that this notion of structure preservation has nothing to do with intrinsic, low-dimensional manifold structure in a dataset, nor does it have anything to do with class structure. We want to preserve the literal, point-cloud metric structure of the latent distribution as exactly as possible, since this implies that downstream representation quality on the original distribution is maintained. Also, we note that our first loss, Markov-Wasserstein minimization (see Section 3.3.1), is only defined for encoders $E : Z \to Z$ where the codomain is the same as the domain, and we take $V = \mathrm{id}_Z$. Wasserstein correlation maximization (Section 3.3.2) additionally allows for dimensionality reduction ($W$ different from $Z$), and we then take $V : Z \to W$ to be an approximate local isometry. In fact, the Wasserstein correlation loss can also be defined without a pretrained network $F$ in the mix at all. In this case, we will say that $E : X \to Z$ is $(\mathbf{t}, \mu_{\mathbf{X}}, \mathbf{V})$-invariant if $E$ is $(t, \mu_X)$-invariant and if there is some $V \in \mathcal{V}(X, Z)$ with $E(x) = V(x)$ on $supp(\mu_X)$.

Finally, we extend Definitions 3.3 and 3.4 to collections of augmentations and to augmented encoders as follows. Let $T = \{t_1, \ldots, t_m\}$ be a collection of augmentations $t_j : A_j \times X \to X$ on $X$ with parameter distributions $\nu_j \in \mathcal{P}_p(A_j)$. An encoder $E : X \to Z$ is $(T, \mu_X)$-invariant if it is $(t_j, \mu_X)$-invariant for each $t_j \in T$. We extend the definitions for $(T, \mu_X, F, V)$-invariance and $(T, \mu_X, V)$-invariance in a similar fashion.

For the case of augmented encoders, we say that $E^T : X \to \mathcal{P}_p(Z)$ is $(T, \mu_X)$-invariant if the underlying deterministic encoder $E : X \to Z$ is $(T, \mu_X)$-invariant, which, in turn, implies that $E^T(x) = \delta_{E(x)}$. Similarly, we say that the augmented encoder $(E \circ F)^T : X \to \mathcal{P}_p(W)$ for $E : Z \to W$ deterministic is $(T, \mu_X, F, V)$-invariant if $E \circ F$ is, which implies that $(E \circ F)^T(x) = \delta_{(V \circ F)(x)}$.

### 3.3 Loss Functions

We now introduce two loss functions for the notions of augmentation invariance defined above, namely, the anchored mean-squared-error (MSE), or Markov-Wasserstein minimization, loss and the Wasserstein correlation maximization loss.

#### 3.3.1 Markov-Wasserstein Minimization (MaWa)

The practical take-away of our work is that, when augmentations have a low collision rate in the pretrained latent space (Definition 3.2), a simple anchored mean-squared error loss is a highly effective objective for post-training augmentation invariance. Let $F : X \to Z$ be a frozen, pretrained network, let $E_\theta : Z \to Z$ be a parameterized encoder, and let $t$ be a (possibly composite) augmentation. Given a batch $\{x_i\}_{i=1}^N$ and $s$ augmentation samples per input, the **Markov-Wasserstein minimization (MaWa)** loss is defined by

$$\theta_* = \arg\min_\theta \mathcal{L}_{\mathrm{MaWa}}(\theta) \coloneqq \frac{1}{N} \sum_{i=1}^N \left( \frac{1}{s+1} \|(E_\theta \circ F)(x_i) - F(x_i)\|_2^2 \right. \tag{21}$$
$$\left. + \frac{1}{s+1} \sum_{k=1}^s \|(E_\theta \circ F)\big(t(a_i^k, x_i)\big) - F(x_i)\|_2^2 \right).$$

The first term enforces preservation of $F$ (anchoring the encoder to the identity on non-augmented inputs) and the second term enforces augmentation invariance (pulling the encoded augmented features toward the corresponding clean features). If all augmentations are collision free, then (21) reaches a minimum of zero when $E_\theta = E_*$ for a $(T, \mu_X, F, \mathrm{id}_Z)$-invariant encoder $E_* : Z \to Z$, since both terms zero out by definition.

This loss avoids the collapse problem that plagues a naive mean squared invariance loss involving terms of the form $\|E_\theta\big(t(a,x)\big) - E_\theta(x)\|_2^2$, which can immediately be minimized by mapping everything in the latent space to a single point. The frozen network $F$ acts similarly to dual network designs, serving as an analog to, say, a slowly updating teacher network or to a stop-gradient protected branch of a twin network.

Below, we show that this loss arises naturally as a minimization problem in the Markov-Wasserstein metric from Definition 2.1. This derivation is not needed to apply the method in practice but provides the formal connection to optimal transport.

**Derivation as Markov-Wasserstein minimization.** Interpreting the equality $(E \circ F)^T = \delta_F$ as an equality of Markov-Wasserstein kernels, we seek

$$\theta_* = \arg\min_\theta \mathrm{MW}_2^2\big((E_\theta \circ F)^T, \delta_F\big). \tag{22}$$

To preserve the metric structure of the pretrained latent space exactly, we include the identity $\mathrm{id}_X$ as a trivial augmentation and set $T = \{\mathrm{id}_X, t\}$ with weights $w = \left(\frac{1}{s+1}, \frac{s}{s+1}\right)$, using the notation from Section 3.2. Then, we can approximate $(E_\theta \circ F)^T$ empirically at $x$ using $s$ samples:

$$(E_\theta \circ F)_s^T(x) = \frac{1}{s+1}\delta_{(E_\theta \circ F)(x)} + \frac{1}{s+1}\sum_{k=1}^s \delta_{(E_\theta \circ F)\big(t(a_x^k,x)\big)}. \tag{23}$$

Previously, we used the notation $a_j^k$ to indicate dependence of the augmentation parameters on the $j$th augmentation. Here, with only one augmentation, we use the notation $a_x^k$ to denote the fact that the augmentation parameters are not necessarily uniform across $\mathrm{supp}(\mu_X)$, but instead come from taking independent samples of $E^T(x)$. When we have a batch $\{x_i\}_{i=1}^N$ with $\mu_X \approx \frac{1}{N}\sum_{i=1}^N \delta_{x_i}$, we will instead write this dependence as $a_i^k$. Note also that

$$(E_\theta \circ F)_s^T(x) \otimes \delta_{F(x)} = \frac{1}{s+1}\delta_{(E_\theta \circ F)(x)} \otimes \delta_{F(x)} + \frac{1}{s+1}\sum_{k=1}^s \delta_{(E_\theta \circ F)(x)} \otimes \delta_{F(x)}. \tag{24}$$

Now, take $p = 2$ and assume that all spaces are standard Euclidean spaces. We use equality (24) together with the fact that couplings with Dirac masses are unique to reduce the Wasserstein computation to a collection of squared $L^2$ terms:

$$MW_2^2\big((E_\theta \circ F)_s^T, \delta_F\big) \tag{25}$$
$$= \int_X W_2^2\big((E_\theta \circ F)_s^T(x), \delta_{F(x)}\big)\mu(dx)$$
$$= \int_X \left(\int_{Z \times Z} \|z_1 - z_2\|_2^2\big((E_\theta \circ F)_s^T(x) \otimes \delta_{F(x)}\big)(dz_1, dz_2)\right)\mu(dx)$$
$$= \int_X \left(\frac{1}{s+1}\|(E_\theta \circ F)(x) - F(x)\|_2^2\right.$$
$$\left. + \frac{1}{s+1}\sum_{k=1}^s \|(E_\theta \circ F)\big(t(a_x^k,x)\big) - F(x)\|_2^2\right)\mu(dx)$$
$$\approx \frac{1}{N}\sum_{i=1}^N \left(\frac{1}{s+1}\|(E_\theta \circ F)(x_i) - F(x_i)\|_2^2\right.$$
$$\left. + \frac{1}{s+1}\sum_{k=1}^s \|(E_\theta \circ F)\big(t(a_i^k,x_i)\big) - F(x_i)\|_2^2\right)$$

recovering the batch loss (21).

### 3.3.2 Wasserstein Correlation Maximization (WaCo)

We now present the **Wasserstein correlation maximization (WaCo)** loss, which achieves invariance results similar to the Markov-Wasserstein minimization framework but that has additional, unique properties. Rather than operate as a loss between networks viewed as elements of a Markov-Wasserstein space, WaCo maximization instead acts on the joint distribution induced by the augmented encoder. Further, besides invariance learning, Wasserstein correlation maximization can alternatively, or simultaneously, be used for dimensionality reduction, with or without the presence of a frozen, pretrained network $F$.

**Remark 3.5** *Crucially, we note that Wasserstein correlation maximization, by itself, is not suitable as a wholly general representation learning method comparable to, say, contrastive methods. The issue is that, by being an approximate (local) isometry on the input distribution, Wasserstein correlation maximization fundamentally cannot cluster points according to semantic similarity in the way that a contrastive method does. On the other hand, this is precisely what makes it suitable for post-training augmentation invariance, whereas other methods fail outright by altering the (metric) structure of the latent space, as demonstrated empirically in Section 4.*

Our setup is as follows. Let $\mu_X \in \mathcal{P}_p(X)$ be a distribution. As before, we focus on the case of a single, possibly composite augmentation $t$, and we again include the identity as a trivial augmentation. That is, we take $T = \{\mathrm{id}_X, t\}$ with weights $w = \left(\frac{1}{s+1}, \frac{s}{s+1}\right)$. We note in passing that, whereas including the identity in the Markov-Wasserstein minimization loss is necessary for strong experimental results, we found that it can be left out of the Wasserstein correlation maximization loss without much degradation. Still, we include it by default.

Markov-Wasserstein minimization requires a frozen, pretrained network $F : X \to Z$ to match against, but for Wasserstein correlation maximization, we have two options. Let $E_\theta : X \to Z$ be a parameterized collection of deterministic encoders. Then, we define the ordinary Wasserstein correlation maximization loss by

$$\theta_* = \arg\max_\theta \mathcal{L}_{\mathrm{WaCo}}(\theta) \coloneqq \mathrm{WC}_p\left(\int_X \left(\delta_x \otimes E_\theta^T(x)\right)\mu_X(dx)\right). \tag{26}$$

That is, we maximize the Wasserstein correlation of the joint distribution induced by the augmented encoder, and there is no requirement that the left and right marginals of this joint distribution live in the same space. (Indeed, recall from Definition 2.3 that all Wasserstein computations here take place in $\mathcal{P}_p(X \times Z)$.) This can be viewed as an objective for $(t, \mu_X, V)$-invariance. Additionally, if $T$ is only the identity, then this loss serves purely to train an encoder that reduces dimensionality while acting as an approximate local isometry on the support of $\mu_X$.

**Remark 3.6** *Although the intuition for the MaWa loss is clear, the WaCo loss is more subtle. We leave a theoretical investigation of the WaCo loss for future work, but the general mechanism is related to the following. First, Theorem 2.2 in Wiesel (2022) shows that the transport-based correlation measure defined there is equal to one if and only if the second marginal is the pushforward of the first marginal under a measurable function. Section 5 of Nies et al. (2021) gives similar results for a variety of transport-based correlation measures. While these results do not apply verbatim to our setting, the idea of using Wasserstein correlation maximization for augmentation invariance is this: In order for the joint distribution induced by the augmented encoder to be far away, in the Wasserstein distance, from the product distribution, subject to normalization constraints, the encoder must bring any augmented views of the same input closer together in the latent space so that the resulting joint distribution is as close as possible to the graph of a measurable function. That is, the encoder must become invariant to the augmentations, and, in practice, we observe that it can accomplish this without representational collapse.*

For the second case of post-training augmentation invariance for a frozen, pretrained network $F : X \to Z$, the loss can be written as

$$\theta_* = \arg\max_\theta \mathcal{L}_{\mathrm{WaCo}}^F(\theta) \coloneqq \mathrm{WC}_p\left(\int_X \left(\delta_{F(x)} \otimes (E_\theta \circ F)^T(x)\right)\mu_X(dx)\right), \tag{27}$$

where $E_\theta : Z \to W$ with $W$ not necessarily the same as $Z$. That is, computations now take place in $\mathcal{P}_p(Z \times W)$.

Our computational implementations of these losses use sliced Wasserstein distances on empirical distributions.[1] We can approximate the induced joint distribution of a batch $\{x_i\}_{i=1}^N$ with $s$ samples as

$$P_N^s\big(X, E_\theta^T(X)\big) = \frac{1}{(s+1)N} \sum_{i=1}^N \delta_{\big(x_i, E_\theta(x_i)\big)} + \frac{1}{(s+1)N} \sum_{i=1}^N \sum_{k=1}^s \delta_{\big(x_i, E_\theta(t(a_i^k, x_i))\big)}. \tag{28}$$

Then, in place of the ordinary WaCo loss (26), we have

$$\theta_* = \arg\max_\theta \mathcal{L}_{\text{SWaCo}}(\theta) \coloneqq \text{SC}_2\Big(P_N^s\big(X, E_\theta^T(X)\big)\Big). \tag{29}$$

Likewise, we define the corresponding sliced version of (27) in the obvious way. Again, our experiments for Wasserstein correlation maximization specifically use these sliced analogs, but, for simplicity, we retain the WaCo naming, instead of switching to SWaCo.

Before proceeding to our experimental results, we make two last remarks. First, when used for dimensionality reduction, the WaCo loss can be expanded to include a reconstruction term for a decoder. The natural loss, at the level of distributions, is given by

$$\mathcal{L}_{\text{dist}}(\phi, \theta) = \alpha W_p\big(\mu_X, (D_\phi \circ E_\theta)_\sharp \mu_X\big) - \beta \text{WC}_p\left( \int_X \big(\delta_x \otimes E_\theta^T(x)\big) \mu_X(dx) \right), \tag{30}$$

and we can formulate a similar loss when reducing and reconstructing from a pretrained latent distribution $F_\sharp \mu_X$. When training with a decoder, we follow Bousquet et al. (2017) and Tolstikhin et al. (2017) and observe that

$$W_c\big(\mu_X, (D_\phi \circ E_\theta)_\sharp \mu_X\big) \le \int_X c\big(x, (D_\phi \circ E_\theta)(x)\big) \mu_X(dx) \tag{31}$$

for any lower semicontinuous cost function $c$. We substitute this upper bound for the reconstruction error, as this allows us to compute a simple $L^2$ reconstruction term in the Euclidean case when $p = 2$. The batch loss is then written as

$$\mathcal{L}(\phi, \theta) = \alpha \frac{1}{N} \sum_{j=1}^N \big\| x_j - (D_\phi \circ E_\theta)(x_j) \big\|_2^2 - \beta \text{SC}_2\Big( P_N^s\big(X, E_\theta^T(X)\big) \Big). \tag{32}$$

Finally, we note that in order for the trained model to be simultaneously invariant to multiple augmentations, one must distinguish between composite versus non-composite augmentations. In general, training for invariance to augmentations $T = \{t_1, \dots, t_m\}$ will only make the encoder separately invariant to each $t_j$. For example, if $t_1$ is rotation and $t_2$ is translation, then the encoder, at least when restricted to simple architectures, will not yet be invariant to inputs that have been both rotated and translated, as given by either $t = t_2 \circ t_1$ or $t = t_1 \circ t_2$. However, if invariance to non-composite augmentations is sufficient, then instead of working with (30) for $T = \{t_1, \dots, t_m\}$, one can also consider taking the logarithm of the product of Wasserstein correlation scores—one for each augmentation $t_j$ and the corresponding augmented encoder—since this allows for parallel computation and may be more efficient, depending on the use case.

## 4 Experiments

To isolate the effect of different training objectives, we employ one-hidden-layer MLPs with ReLU activations, deliberately avoiding more complex architectures that involve, for example, convolution, attention, or

---

[1]While it is well known that sliced distances are an effective approximation of ordinary Wasserstein distances (see, e.g., Bonnotte (2013) or Nadjahi et al. (2019)), we note that to have a fully rigorous connection between sliced Wasserstein correlation (SWC) and ordinary Wasserstein correlation (WC), we would need to justify that SWC maximization is indeed a suitable proxy for WC maximization, similar to how InfoNCE losses can be shown to be a lower bound on mutual information maximization losses. This is left for future work. In any case, we demonstrate empirically that, if nothing else, SWC maximization does indeed serve as a suitable objective for (post-training) augmentation invariance.

normalization layers. (Exact details can be found in Appendix B.) This approach better ensures that the structure preservation and invariance properties in the trained models are not the result of inductive biases in the networks themselves. It also shows that the method is extremely lightweight and can be used to modify pretrained networks with little additional training burden.

## 4.1 Evaluation Framework

Our tests are designed to evaluate $(t, \mu_X, F, V)$-invariance for various choices of $F$ and $V$. We use five pretrained networks spanning different architectures and training objectives:

- **DINO**: a vision transformer[2] with $d = 384$ trained on the self-distillation with no labels (DINO) objective (Caron et al., 2021).
- **SwAV**: a ResNet50[3] with $d = 2048$ trained on the Swapping Assignments between Views (SwAV) objective (Caron et al., 2020).
- **R-DINO**: a ResNet50[4] with $d = 2048$ trained on the DINO objective, providing a direct comparison of backbone architectures under the same training objective.
- **CLIP**: a vision transformer[5] with $d = 512$ trained on the contrastive language–image pretraining objective (Radford et al., 2021).
- **ResNet50**: a supervised ResNet50[6] with $d = 2048$ trained on ImageNet classification, included to test whether our methods extend beyond self-supervised feature spaces.

For our primary experiments, we use the STL10 dataset[7] with ImageNet normalization statistics. To evaluate scaling behavior with respect to the number of classes, we provide additional tests for DINO on TinyImageNet (200 classes) and on a 50-class subset of TinyImageNet selected via a greedy procedure to maximize pairwise class separation in the DINO latent space. Partial results for the primary DINO and SwAV experiments are reported in the main text, with all remaining results reported in Appendix C. In addition to our results for the MaWa minimization loss and the WaCo maximization loss, we compare against two alternative ways one might try to achieve $(t, \mu_X, F, V)$-invariance. Specifically, we use SimCLR (Chen et al., 2020) (denoted by SCLR in our tables for more uniform presentation) as a representative contrastive method, and we use maximization of the Hilbert-Schmidt Independence Criterion (HSIC) as a possible alternative to the WaCo loss.

The SimCLR loss pushes augmented views closer together, which is desirable for invariance, but, as will be seen, when training an encoder $E_\theta$ in this way on augmented views of the form $F\big(t(a,x)\big)$ for a pretrained network $F$, the original latent space of $F$ can become badly corrupted, breaking one of our main requirements for post-training augmentation invariance.

For HSIC comparisons, we use the $O(1/N)$ biased estimator from Gretton et al. (2005), which was shown to be an effective loss for representation learning in Li et al. (2021). That is, given i.i.d. samples $\{(x_i, y_i)\}_{i=1}^N$ from a joint distribution $(X, Y)$,

$$\mathrm{HSIC}(X, Y) = \frac{1}{(N-1)^2} \mathrm{Tr}(KHLH) \tag{33}$$

for kernel matrices $K_{ij} = k(x_i, x_j)$, $L_{ij} = l(y_i, y_j)$ and centering matrix $H = I - \frac{1}{N}\mathbf{1}\mathbf{1}^T$. Specifically, we train $E_\theta$ using HSIC maximization on the joint distribution $P_N^s\big(X, E_\theta^T(X)\big)$ induced by the augmented encoder. However, as is the case with SimCLR, we'll see that this potential alternative is also ill-suited for post-training augmentation invariance, since the latent space of $F$ can again become badly corrupted.

---

[2]The dino_vits8 model available at `https://github.com/facebookresearch/dino`

[3]Available at `https://github.com/facebookresearch/swav`

[4]The dino_resnet50 model again available at `https://github.com/facebookresearch/dino`

[5]The CLIP ViT-B/16 image encoder available at `https://github.com/openai/CLIP.git`

[6]Available through Torchvision

[7]STL10, we submit, provides a reasonable compromise between realistic feature processing through large pretrained networks and computational efficiency.

### 4.1.1 Invariance Evaluation

To evaluate invariance, we compare classification accuracy of $C \circ E_\theta \circ F$ versus $C \circ F$ on augmented data for a classifier $C$ trained on non-augmented data. In the first case, for $C \circ E_\theta \circ F$, $C$ is either a linear classifier (LC), or else a nonlinear classifier (NC), specifically a simple, one-hidden-layer MLP with ReLU activation. We call $C \circ F$ the end-to-end classifier (EC) case, and $C$ in this case is a nonlinear MLP with the same total depth as $C \circ E_\theta$ for $C$ linear. In all cases, we train on a standard cross-entropy loss for 50 epochs. It is certainly possible to obtain better classification results with longer training times and more fine-tuning, but we are primarily concerned with comparing relative accuracy between networks of the form $C \circ E_\theta \circ F$ and $C \circ F$.

Classification is a more practical test of invariance than direct measurement of whether we have (approximate) equalities $E_\theta\big(t(a,x)\big) = E_\theta(x)$, since in pretrained latent spaces, raw $L2$ distances between points in the same class can actually be quite large, making these calculations uninformative in practice. Classification accuracy instead shows in a task-relevant way that the augmented data lives within the same decision boundaries as the non-augmented data after passing through $E_\theta$.

Finally, all augmentations are taken from the Kornia package (Riba et al., 2020). We use (arbitrary, 360 degree) rotations, affine transformations, noise, and crops as our augmentation types. Full experimental details and Kornia parameters can be found in Appendix B.

**Remark 4.1** *What exactly is different from our approach to simply training a classifier on augmented data directly? A classifier $C$ trained on data of the form $\{(F\big(t(a,x_i)\big), label(x_i)\}_{i=1}^{N}$ would indeed be an invariant classifier, but the latent space of $F$ itself need not be invariant. Indeed, under natural assumptions, we can have $(C \circ F)\big(t(a,x)\big) = (C \circ F)(x)$, but $F(t(a,x))$ need not equal $F(x)$, even approximately. If one only wants an invariant classifier going into logit space $C : Z \to \mathbb{R}^m$ for $m$ classes, then this is acceptable. However, the advantage of having an adapter $E_\theta$ that turns $E_\theta \circ F$ into a $(t, \mu_X, F, V)$-invariant encoder is that one then obtains invariance for any downstream task, not just classification. Again, our aim is to alter the latent space of $F$ itself without degrading its existing performance. This is why we train our classifiers on non-augmented data and then evaluate on augmented data: to show that the latent space itself has been reshaped. In particular, evaluating linear probes on the latent space is a standard evaluation protocol in (self-supervised) representation learning.*

### 4.1.2 Structure Evaluation

To evaluate whether or not the original latent space is preserved, we test to what degree $E_\theta$ acts isometrically on $F_\sharp \text{supp}(\mu_X)$. Specifically, we consider scatter plots of the form

$$\big\{(\|F(x) - F(y)\|_2, \|(E_\theta \circ F)(x) - (E_\theta \circ F)(y)\|_2)\big\}_{x,y \in \text{supp}(\mu_X)}. \tag{34}$$

That is, we plot $\|(E_\theta \circ F)(x) - (E_\theta \circ F)(y)\|_2$ against $\|F(x) - F(y)\|_2$. If $E_\theta$ is exactly isometric on the original, non-augmented features, then all points will lie on the line $y = x$. Naturally, we don't expect a perfect isometry in practice, so we record several complementary statistics that together characterize the quality of the approximation.

First, we record the lower and upper Lipschitz constants $L_1$ and $L_2$, defined by

$$L_1 := \min_{\substack{x,y \in \text{supp}(\mu_X) \\ x \neq y}} \frac{\|(E_\theta \circ F)(x) - (E_\theta \circ F)(y)\|_2}{\|F(x) - F(y)\|_2}, \text{ and} \tag{35}$$

$$L_2 := \max_{\substack{x,y \in \text{supp}(\mu_X) \\ x \neq y}} \frac{\|(E_\theta \circ F)(x) - (E_\theta \circ F)(y)\|_2}{\|F(x) - F(y)\|_2}.$$

For a perfect isometry, $L_1 = L_2 = 1$.

Second, we record the slope $m$ and intercept $b$ of the ordinary least-squares fit to (34), together with the coefficient of determination

$$R^2 = 1 - \frac{\text{SS}_{\text{res}}}{\text{SS}_{\text{tot}}}, \tag{36}$$

where $\mathrm{SS}_{\mathrm{res}}$ is the sum of squared residuals of the fitted line and $\mathrm{SS}_{\mathrm{tot}}$ is the total sum of squares. These quantities characterize the best affine approximation to the relationship between input and encoded distances and how tightly the data concentrates around it.

Third, to directly quantify deviation from isometric behavior, we report the root-mean-square deviation (RMSD) of the encoded distances from the input distances. Writing $d_{xy} = \|F(x) - F(y)\|_2$ and $e_{xy} = \|(E_\theta \circ F)(x) - (E_\theta \circ F)(y)\|_2$ for distinct pairs $x, y \in \mathrm{supp}(\mu_X)$, RMSD is defined by

$$\mathrm{RMSD} = \sqrt{\frac{1}{N} \sum_{x \neq y} (e_{xy} - d_{xy})^2}, \tag{37}$$

where $N$ is the number of distinct pairs. This measures the typical absolute deviation from the isometry line $y = x$ in the same units as the pairwise distances. We additionally report two scale-free normalizations that facilitate comparison across different pretrained networks and latent space dimensions: the normalized RMSD (NRMSD), defined as

$$\mathrm{NRMSD} = \frac{\mathrm{RMSD}}{d_{\max} - d_{\min}}, \tag{38}$$

where $d_{\max}$ and $d_{\min}$ are the maximum and minimum values of $\{d_{xy}\}$, and the coefficient of variation of the RMSD (CVRMSD), defined as

$$\mathrm{CVRMSD} = \frac{\mathrm{RMSD}}{\bar{d}}, \tag{39}$$

where $\bar{d}$ is the mean input distance. Both normalized quantities equal zero for a perfect isometry. NRMSD expresses the typical deviation as a fraction of the total range of input distances, while CVRMSD expresses it as a fraction of the mean input distance.

These quantities provide the most direct evaluation for the structure-preservation constraint demanded by $(t, \mu_X, F, V)$-invariance where $V$ is a (local, approximate) isometry. That said, our evaluation code also contains optional computations of (normalized) spectral statistics and heat kernel statistics of $k$-nn feature graphs for a range of possible values of $k$. These tests are meant to further evaluate how well the encoder $E_\theta$ preserves local geometric properties of the latent space, and we found that our trained encoders consistently performed well. Overall, though, these tests were not as informative or illuminating as the direct isometry testing outlined above, so we leave them out to streamline our presentation.

Finally, we stress once more that we are interested in pretrained features taken as a point cloud in a Euclidean space. We are not concerned with structure in the sense of low-dimensional intrinsic manifolds or class structure. If the metric structure of the original pretrained latent space is perfectly preserved, then downstream tasks on the original distribution will continue to behave exactly as expected. More generally, for any downstream task involving, say, linear probes on the latent space, more or less distortion, which is to say, more or less rigidity for the choice of $V$, may be tolerated.

## 4.2 Results

### 4.2.1 DINO (Transformer Backbone)

Table 1 shows classification accuracy on both original and augmented data for the pretrained network $F = \mathrm{DINO}$ and for one-hidden-layer MLP encoders $E_\theta$ trained with the four losses enumerated previously. The first number of each entry reports classification accuracy on non-augmented data, and the second number is accuracy on augmented data. For example, the entry for row (NC, MaWa) and column Rotation, namely, the pair 98.62 | 94.74, should be read as follows: The composite network $C \circ E_\theta \circ F$ (for a nonlinear classifier $C$ trained on non-augmented data and for $E_\theta$ trained for rotation-invariance with MaWa minimization) achieves 98.62% classification accuracy on non-rotated STL10 DINO features (the first number). Further, on arbitrarily rotated images processed through $F$, $C \circ E_\theta \circ F$ achieves 94.74% accuracy (the second number). This is in contrast to the EC, or end-to-end classifier, row, which shows that the network $C \circ F$ (for $C$ nonlinear with total depth equal to $C \circ E_\theta$) drops to 71.22% accuracy for arbitrarily rotated images.

Across augmentation types, the MaWa minimization loss and the WaCo maximization loss perform strongly, with MaWa consistently achieving the best results. The SimCLR and HSIC losses are not competitive

Table 1: Classification comparison across models for linear (LC), nonlinear (NC), and end-to-end (EC) classifiers on STL10 with DINO features. First number is accuracy on original data; second is accuracy on augmented data.

| Augmentation | | Rotation | Affine | Noise | Crop |
|---|---|---|---|---|---|
| **LC** | MaWa | 98.22 \| 94.66 | 98.19 \| 97.20 | 98.11 \| 86.09 | 98.20 \| 98.11 |
| | WaCo | 97.66 \| 91.09 | 98.19 \| 96.33 | 97.26 \| 78.78 | 97.97 \| 97.66 |
| | SCLR | 91.21 \| 86.30 | 97.40 \| 95.90 | 94.24 \| 44.47 | 97.85 \| 97.60 |
| | HSIC | 77.38 \| 72.98 | 96.73 \| 90.64 | 95.55 \| 45.47 | 95.06 \| 93.93 |
| **NC** | MaWa | 98.62 \| **94.74** | 98.65 \| **97.43** | 98.55 \| **86.23** | 98.66 \| **98.36** |
| | WaCo | 98.02 \| 92.30 | 98.42 \| 96.83 | 97.71 \| 80.75 | 98.32 \| 98.00 |
| | SCLR | 91.38 \| 86.72 | 97.41 \| 95.71 | 95.40 \| 57.75 | 97.92 \| 97.64 |
| | HSIC | 87.42 \| 81.28 | 97.24 \| 92.10 | 97.02 \| 45.78 | 96.56 \| 95.76 |
| **EC** | | 98.58 \| 71.22 | 98.58 \| 94.65 | 98.58 \| 58.22 | 98.58 \| 98.10 |

on rotations and noise augmentations. The success of the EC case for affine and crop augmentations, as well as the stronger results for SimCLR and HSIC on these augmentation types, is expected: The DINO model is already strongly (approximately) invariant to these augmentations, since crops and moderate affine transformations are central to how the model was trained. Given this, we focus going forward on the results for rotation and noise augmentations where the gap between the adapter-equipped networks and the baseline is most pronounced.

Importantly, the MaWa and WaCo losses consistently preserve the structure of the original latent space, whereas SimCLR and HSIC introduce major corruptions. Tables 2 and 3 show that MaWa minimization achieves near-isometric behavior on the latent distribution: For rotations, $R^2 = 0.96$, NRMSD = CVRMSD = 0.02, and the Lipschitz constants are tightly bounded ($L_1 = 0.77$, $L_2 = 1.11$). The WaCo loss preserves the latent space less tightly ($R^2 = 0.55$ at full dimension), though it still yields competitive classification results and additionally supports dimensionality reduction. Reducing from $d = 384$ to $d = 96$ maintains comparable classification accuracy while slightly improving the $R^2$ of the linear fit.

Table 2: Structure preservation for MaWa and WaCo rotation-invariant encoders on DINO features.

| Structure | | STL10 + DINO + Rotation | | |
|---|---|---|---|---|
| Preservation | | MaWa ($d = 384$) | WaCo ($d = 384$) | WaCo ($d = 96$) |
| Similarity | $L_1 \mid L_2$ | 0.77 \| 1.11 | 0.48 \| 1.62 | 0.48 \| 1.78 |
| Tests | $m \mid b$ | 0.99 \| 0.43 | 1.11 \| -15.80 | 1.18 \| -25.00 |
| | $R^2$ | 0.96 | 0.55 | 0.66 |
| | RMSD | 1.78 | 9.16 | 9.00 |
| | NRMSD | 0.02 | 0.09 | 0.09 |
| | CVRMSD | 0.02 | 0.08 | 0.08 |
| Classification | LC | 98.22 \| 94.66 | 97.66 \| 91.09 | 97.28 \| 91.19 |
| | NC | 98.62 \| 94.74 | 98.02 \| 92.30 | 98.02 \| 92.82 |
| | EC | 98.58 \| 71.22 | 98.58 \| 71.22 | 98.58 \| 71.22 |

The results for noise augmentations reported in Table 3 are similar. The drop in accuracy is more pronounced, since noise is a highly destructive, non-invertible augmentation, but the MaWa loss still yields a nearly 28 percentage point increase in accuracy over the EC baseline (86.23% versus 58.22% in the NC case). Further, structure preservation remains strong, with $R^2 = 0.98$ and NRMSD = CVRMSD = 0.01 for the MaWa loss.

Table 3: Structure preservation for MaWa and WaCo noise-invariant encoders on DINO features.

| Structure | | STL10 + DINO + Noise | | |
|---|---|---|---|---|
| Preservation | | MaWa ($d = 384$) | WaCo ($d = 384$) | WaCo ($d = 96$) |
| Similarity | $L_1 \mid L_2$ | 0.87 \| 1.13 | 0.47 \| 1.65 | 0.47 \| 1.59 |
| Tests | $m \mid b$ | 1.00 \| 0.24 | 1.11 \| -14.05 | 1.18 \| -25.55 |
| | $R^2$ | 0.98 | 0.52 | 0.64 |
| | RMSD | 1.25 | 9.10 | 9.52 |
| | NRMSD | 0.01 | 0.09 | 0.09 |
| | CVRMSD | 0.01 | 0.08 | 0.09 |
| Classification | LC | 98.11 \| 86.09 | 97.26 \| 78.78 | 97.20 \| 79.49 |
| | NC | 98.55 \| 86.23 | 97.71 \| 80.75 | 97.58 \| 81.85 |
| | EC | 98.58 \| 58.22 | 98.58 \| 58.22 | 98.58 \| 58.22 |

We also visualize these structure-preservation results directly by plotting the best linear fit for the scatter plot of input and encoded distance pairs. Figures 1 and 2 are for the MaWa and WaCo losses, respectively, in the case of rotation invariance, and we see clearly the additional variance introduced by the WaCo loss. The plots for noise invariance look nearly identical.

In contrast, the structural results for SimCLR and HSIC on rotation and noise augmentations in Tables 4 and 5 show that $R^2$ plummets (to 0.04 and 0.06 for rotations, respectively), while NRMSD and CVRMSD exceed 1.0, indicating that the encoded distances bear little resemblance to the original pairwise distances. Figures 3 and 4 illustrate the contrast directly. The HSIC loss, in particular, leads to near-total collapse of the distance structure.

Table 4: Structure preservation for SimCLR and HSIC rotation-invariant encoders on DINO features.

| Structure | | STL10 + DINO + Rotation | |
|---|---|---|---|
| Preservation | | SCLR ($d = 384$) | HSIC ($d = 384$) |
| Similarity | $L_1 \mid L_2$ | 0.21 \| 13.91 | 0.01 \| 0.33 |
| Tests | $m \mid b$ | 3.68 \| -56.11 | 0.04 \| -0.89 |
| | $R^2$ | 0.04 | 0.06 |
| | RMSD | 286.57 | 108.63 |
| | NRMSD | 2.77 | 1.05 |
| | CVRMSD | 2.57 | 0.97 |
| Classification | LC | 91.21 \| 86.30 | 77.38 \| 72.98 |
| | NC | 91.38 \| 86.72 | 87.42 \| 81.28 |
| | EC | 98.58 \| 71.22 | 98.58 \| 71.22 |

We provide one final comparison of the different methods for the case of $F = $ DINO. Figure 5 shows $t$-SNE visualizations of the original STL10 DINO latent distribution $F_\sharp \mu_X$ compared to $(E_\theta \circ F)_\sharp \mu_X$ for encoders

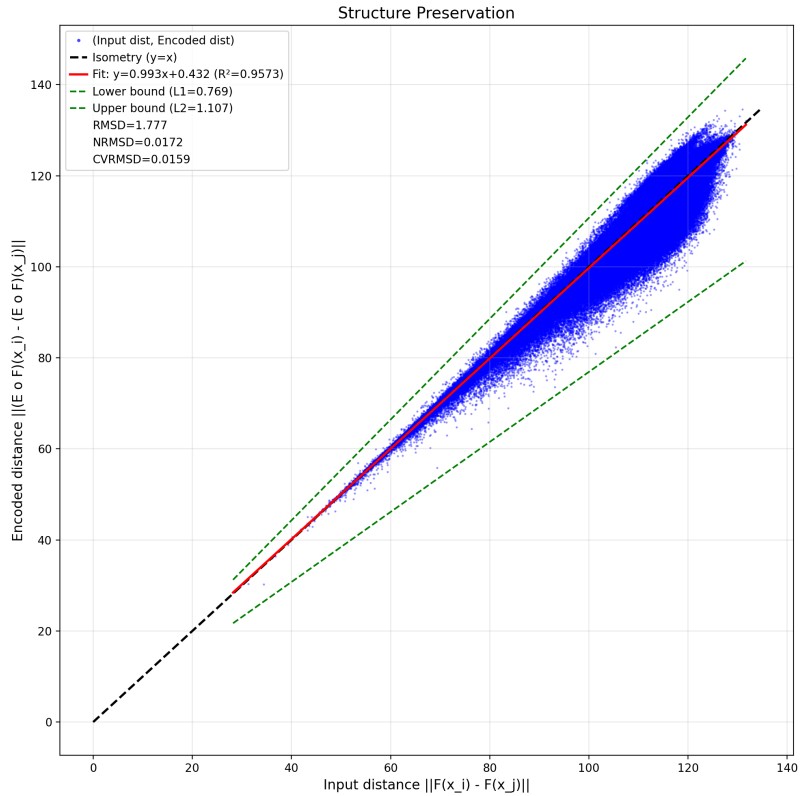

Figure 1: Structure preservation for MaWa rotation-invariant encoder on DINO features.

Table 5: Structure preservation for SimCLR and HSIC noise-invariant encoders on DINO features.

| Structure Preservation | | STL10 + DINO + Noise | |
| --- | --- | --- | --- |
| | | SCLR ($d = 384$) | HSIC ($d = 384$) |
| Similarity Tests | $L_1 \mid L_2$ | 2.09 \| 19.34 | 0.22 \| 12.47 |
| | $m \mid b$ | 11.01 \| -223.70 | 3.12 \| -90.81 |
| | $R^2$ | 0.16 | 0.03 |
| | RMSD | 920.14 | 218.36 |
| | NRMSD | 8.90 | 2.11 |
| | CVRMSD | 8.25 | 1.96 |
| Classification | LC | 94.24 \| 44.47 | 95.55 \| 45.47 |
| | NC | 95.40 \| 57.75 | 97.02 \| 45.78 |
| | EC | 98.58 \| 58.22 | 98.58 \| 58.22 |

trained for noise invariance using one of the four losses. Both optimal transport-based losses strongly preserve the class structure (in virtue of approximately preserving metric structure), whereas SimCLR and HSIC produce noticeable corruption. The $t$-SNE plots for rotation invariance show exactly similar patterns.

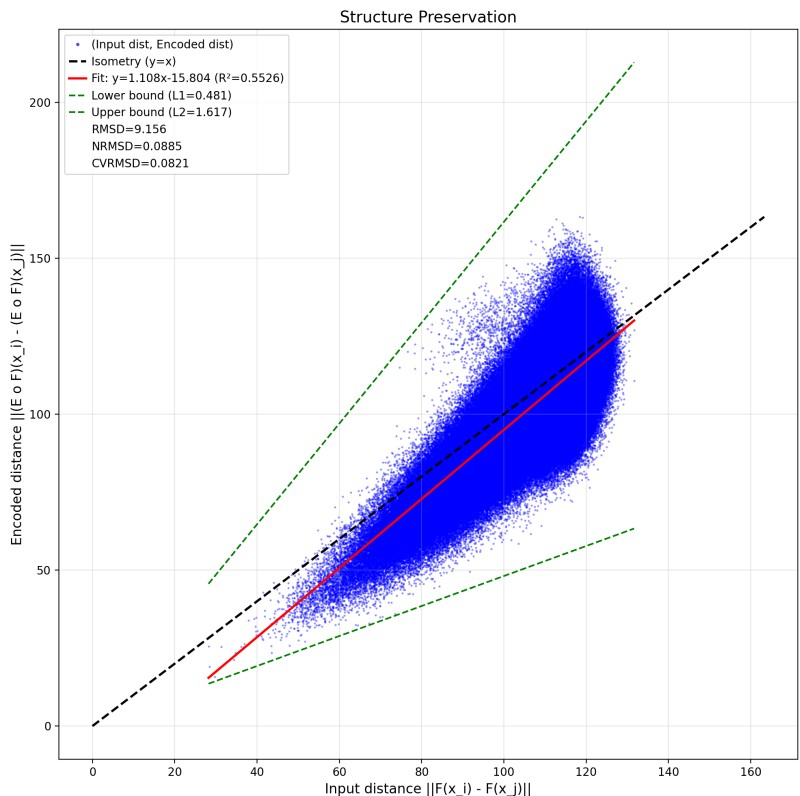

Figure 2: Structure preservation for WaCo rotation-invariant encoder on DINO features.

### 4.2.2 SwAV (ResNet Backbone)

We now illustrate how the choice of pretrained network $F$ affects results. For $F = $ SwAV, Table 6 shows classification results across all four augmentation types, and Tables 7 and 8 report the corresponding structure-preservation metrics for rotation and noise. (We again focus here on rotations and noise, since, as with DINO, SwAV already has stronger baseline invariance to crops and affine transformations.) Invariant classification is not as strong as for the DINO transformer, but the relative increase in accuracy over the EC baseline is actually greater. For noise, for example, the EC classifier is essentially unusable at 40.82% accuracy, whereas the MaWa encoder achieves 73.18% accuracy in the linear case. Structure preservation remains strong for MaWa and WaCo models. Full results for SwAV are reported in Appendix C.

### 4.2.3 Additional Pretrained Networks

To test the generality of our approach, we run the full suite of experiments on three additional pretrained networks: DINO with a ResNet backbone (R-DINO, $d = 2048$), CLIP ($d = 512$), and a supervised ResNet50 ($d = 2048$). We also evaluate DINO on TinyImageNet (200 classes) and a 50-class subset of TinyImageNet selected via a greedy procedure to maximize pairwise class separation in the DINO latent space. Full results for all models, augmentation types, and losses are reported in Appendix C. We summarize the main findings here.

For CLIP and R-DINO on STL10, the MaWa loss continues to perform strongly across all augmentation types. CLIP, in particular, yields the best overall results. On rotations, the MaWa-invariant CLIP network achieves 94.82% accuracy (LC) versus 88.16% for the EC baseline, with near-perfect structure preservation ($R^2 = 0.99$, NRMSD = CVRMSD = 0.01). R-DINO shows a pattern intermediate between DINO and SwAV, showing that both loss and backbone matter. The collision rates for these models (Tables 44 and 45

Table 6: Classification comparison on STL10 with SwAV features.

| Augmentation | | Rotation | Affine | Noise | Crop |
|---|---|---|---|---|---|
| **LC** | MaWa | 93.44 \| 81.58 | 93.70 \| 88.42 | 93.97 \| 73.18 | 93.73 \| 94.32 |
| | WaCo | 88.21 \| 80.86 | 90.09 \| 86.00 | 87.17 \| 68.23 | 90.74 \| 91.39 |
| | SCLR | 79.50 \| 76.31 | 87.56 \| 85.62 | 73.94 \| 46.88 | 92.92 \| 93.34 |
| | HSIC | 71.90 \| 69.49 | 81.24 \| 78.98 | 65.83 \| 59.97 | 87.95 \| 87.94 |
| **NC** | MaWa | 93.21 \| 80.79 | 93.46 \| **87.74** | 93.59 \| **72.84** | 93.58 \| 93.98 |
| | WaCo | 88.59 \| **80.98** | 90.26 \| 86.05 | 86.60 \| 67.70 | 91.14 \| 91.63 |
| | SCLR | 79.69 \| 76.78 | 87.78 \| 86.06 | 76.39 \| 47.06 | 92.33 \| 92.84 |
| | HSIC | 81.67 \| 77.83 | 87.96 \| 85.62 | 74.38 \| 65.05 | 92.67 \| 92.69 |
| **EC** | | 94.23 \| 50.35 | 94.23 \| 76.06 | 94.23 \| 40.82 | 94.23 \| **94.33** |

Table 7: Structure preservation for MaWa and WaCo rotation-invariant encoders on SwAV features.

| Structure Preservation | | STL10 + SwAV + Rotation | | |
|---|---|---|---|---|
| | | MaWa ($d = 2048$) | WaCo ($d = 2048$) | WaCo ($d = 512$) |
| Similarity Tests | $L_1$ \| $L_2$ | 0.44 \| 0.96 | 0.37 \| 1.80 | 0.51 \| 1.43 |
| | $m$ \| $b$ | 0.91 \| -1.56 | 1.72 \| -7.08 | 1.28 \| -3.06 |
| | $R^2$ | 0.87 | 0.81 | 0.84 |
| | RMSD | 2.62 | 2.12 | 1.01 |
| | NRMSD | 0.15 | 0.12 | 0.06 |
| | CVRMSD | 0.23 | 0.18 | 0.09 |
| Classification | LC | 93.44 \| 81.58 | 88.21 \| 80.86 | 89.34 \| 81.52 |
| | NC | 93.21 \| 80.79 | 88.59 \| 80.98 | 90.28 \| 81.41 |
| | EC | 94.23 \| 50.35 | 94.23 \| 50.35 | 94.23 \| 50.35 |

Table 8: Structure preservation for MaWa and WaCo noise-invariant encoders on SwAV features.

| Structure Preservation | | STL10 + SwAV + Noise | | |
|---|---|---|---|---|
| | | MaWa ($d = 2048$) | WaCo ($d = 2048$) | WaCo ($d = 512$) |
| Similarity Tests | $L_1$ \| $L_2$ | 0.58 \| 0.99 | 0.32 \| 2.52 | 0.41 \| 1.63 |
| | $m$ \| $b$ | 0.94 \| -1.11 | 1.76 \| -7.46 | 1.14 \| -1.93 |
| | $R^2$ | 0.95 | 0.76 | 0.77 |
| | RMSD | 1.79 | 2.40 | 1.09 |
| | NRMSD | 0.10 | 0.13 | 0.06 |
| | CVRMSD | 0.15 | 0.21 | 0.09 |
| Classification | LC | 93.97 \| 73.18 | 87.17 \| 68.23 | 88.84 \| 70.38 |
| | NC | 93.59 \| 72.84 | 86.60 \| 67.70 | 89.39 \| 69.41 |
| | EC | 94.23 \| 40.82 | 94.23 \| 40.82 | 94.23 \| 40.82 |

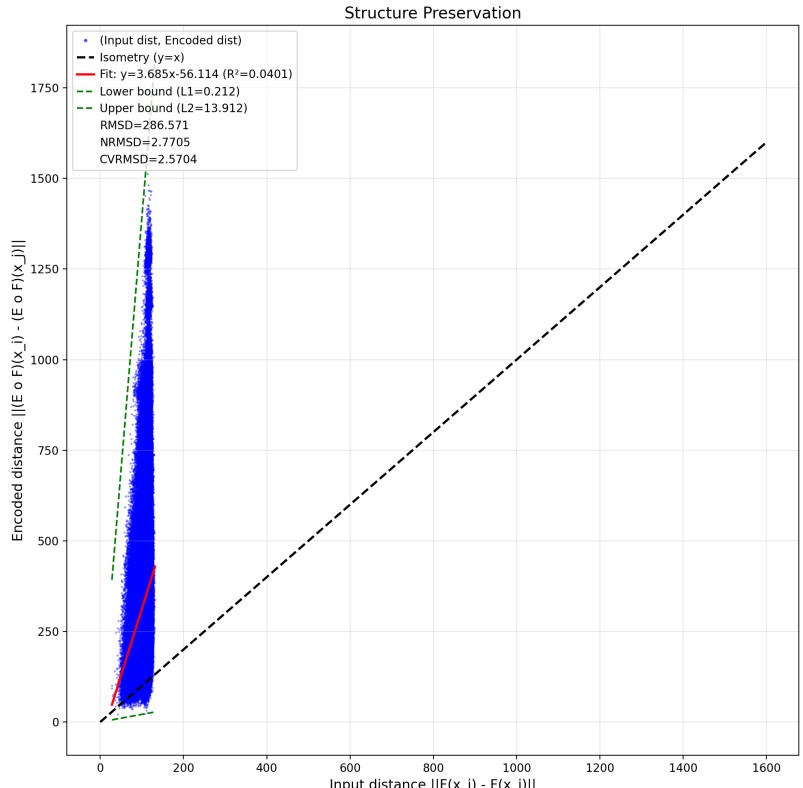

Figure 3: Structure preservation for SimCLR rotation-invariant encoder on DINO features.

in Appendix C) confirm the pattern: both have low aligned collision rates, predicting the success of MaWa. (See 4.2.4 below for more on collisions.)

The supervised ResNet50 presents a qualitatively different case. Here, the MaWa loss collapses altogether. The competing constraints introduced by the high collision rate (see Table 11) prevent the optimizer from finding a meaningful solution, and classification accuracy on even non-augmented data drops to approximately 50%. The WaCo loss, which does not anchor to the identity, is more robust to this failure mode and still produces useful (approximately) invariant encoders on ResNet50, achieving, for example, 70.06% accuracy (LC case) on rotated data versus 48.63% for the EC baseline.

For DINO on TinyImageNet, classification accuracy is naturally lower due to the increased number of classes (200 versus 10), but the relative improvements from the adapter networks remain consistent. On the 50-class TinyImageNet subset, where inter-class separation is larger, the MaWa loss achieves 83.59% rotation-invariant accuracy (LC case) versus 53.17% for the EC baseline.

### 4.2.4 Collision Rate Analysis

The collision rate from Definition 3.2 measures the fraction of augmented samples $F(t(a,x))$ whose nearest clean neighbor in the latent space belongs to a different class. High collision rates indicate that the augmentation has displaced features across class boundaries, creating competing optimization constraints.

Tables 9–11 report collision rates for DINO, SwAV, and ResNet50 on STL10, and Tables 44–47 in Appendix C report collision rates for all remaining tests. We compute two versions: the raw collision rate on augmented features $F(t(a,x))$, and the aligned collision rate, which is the collision rate after aligning the augmented feature cloud to the clean feature cloud via the best rigid isometry (orthogonal transformation plus translation). The difference isolates how much of the collision problem is due to a globally coherent displacement, correctable by a rigid transformation, versus local, residual entanglement.

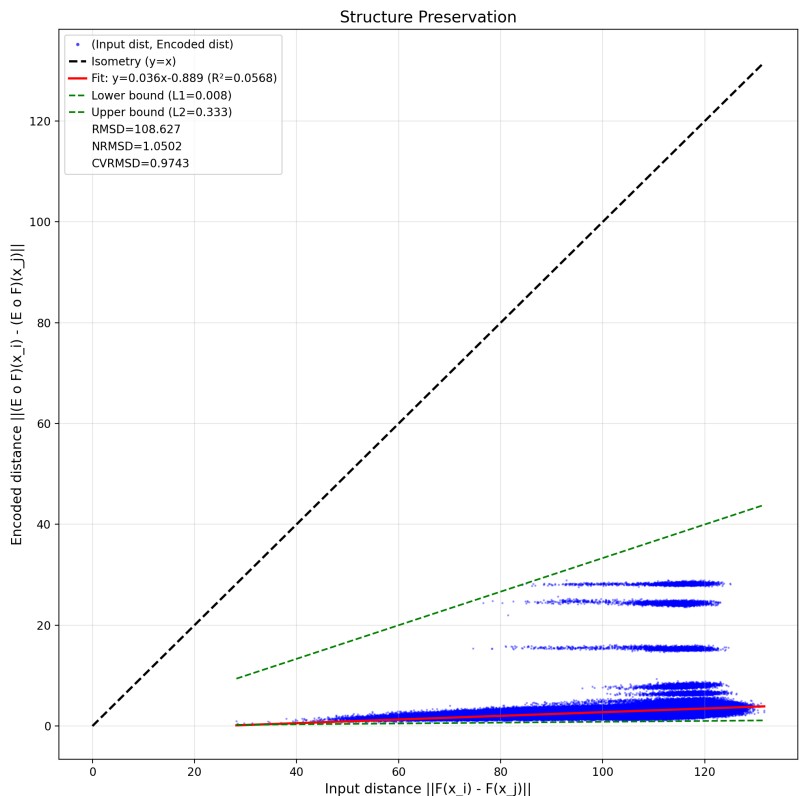

Figure 4: Structure preservation for HSIC rotation-invariant encoder on DINO features.

Several patterns emerge. First, the self-supervised models (DINO, SwAV, R-DINO, CLIP) exhibit dramatically lower aligned collision rates than the supervised ResNet50. This indicates that augmentations act on self-supervised feature spaces in a globally coherent manner. Second, the aligned collision rate roughly predicts the feasibility of MaWa minimization: For models with lower aligned collision rates, the method succeeds, while ResNet50, which retains substantial residual collisions (e.g., 13.1% for rotations and 41.8% for noise after alignment), is the only model on which MaWa produces collapse. The WaCo loss, which does not anchor to the identity, is more robust to this failure mode.

Table 9: Collision rates for DINO features on STL10 before (raw) and after rigid alignment.

| Augmentation | CR (%, raw) | CR (%, aligned) |
|---|---|---|
| Rotation | 9.20 | 0.00 |
| Affine | 0.00 | 0.00 |
| Noise | 22.40 | 0.60 |
| Crop | 0.00 | 0.00 |

### 4.2.5 WaCo for Dimensionality Reduction

We discuss one last test to better illuminate the properties of the WaCo loss, which, as noted, can be defined with or without a pretrained network $F$ and can be used simultaneously for dimensionality reduction, or for dimensionality reduction alone. We train an encoder $E_\theta$ on MNIST using the ordinary WaCo loss (26) with, in the first case, $T = \{\mathrm{id}_X\}$ or with $T = \{\mathrm{id}_X, t\}$ in the second, where $t$ is arbitrary rotations. Figure 6 shows $t$-SNE visualizations of the learned latent spaces for ordinary MNIST digits plus 90-degree rotations.

Table 10: Collision rates for SwAV features on STL10 before (raw) and after rigid alignment.

| Augmentation | CR (%, raw) | CR (%, aligned) |
|---|---|---|
| Rotation | 47.00 | 0.10 |
| Affine | 15.90 | 0.20 |
| Noise | 74.70 | 4.80 |
| Crop | 0.10 | 0.00 |

Table 11: Collision rates for supervised ResNet50 features on STL10 before (raw) and after rigid alignment.

| Augmentation | CR (%, raw) | CR (%, aligned) |
|---|---|---|
| Rotation | 46.40 | 13.10 |
| Affine | 21.00 | 5.10 |
| Noise | 77.30 | 41.80 |
| Crop | 2.10 | 0.50 |

In the first case, the 90-degree rotated digits are encoded as separate classes, whereas in the second case, we see near perfect overlap between the original classes and their rotated counterparts. (We emphasize again that although we visualize the case of 90-degree rotations, we trained on arbitrary rotations.) Finally, we note that the notion of invariance we have developed in this work depends on the choice of the initial distribution $\mu_X$. If $\mu_X$ in this final test were instead 90-degree rotations of MNIST digits, or, say, the ordinary MNIST distribution plus an additional class, then that would be the structure we want to preserve, since augmentation invariance in our sense only applies to unique augmentations, as defined previously.

## 5 Conclusion

We have developed precise definitions of augmented encoders and augmentation invariance with emphasis on the problem of post-training augmentation invariance, in which our goal is to add additional invariance properties to a pretrained network without corrupting its behavior on the original, non-augmented distribution. Our experimental results show that both Markov-Wasserstein minimization and Wasserstein correlation maximization are effective losses for this task across a broad range of settings, whereas SimCLR and HSIC maximization losses were shown to be consistently ineffective. Additionally, we have shown that the proposed methods are lightweight and do not depend on complicated architectures or fine hyperparameter tuning. With the right training objective, post-training augmentation invariance can be achieved, at least approximately, with only a one-hidden-layer MLP appended to the latent space of a frozen, pretrained network.

### 5.1 Limitations

While we have shown that our methods are effective across a range of augmentation types, including affine transformations, which are themselves composite augmentations involving rotation, translation, scaling, and shear, we note that convergence slows as the augmentation parameter space grows. Improving sample efficiency for larger parameter spaces, for example through curriculum-based sampling or more expressive adapter architectures, is left for future work.

Additionally, our collision rate analysis reveals a structural limitation of the MaWa loss: When the pretrained feature space has high residual collision rates after rigid alignment, as is the case for the supervised ResNet50, the loss objective faces too many competing constraints and collapses. The WaCo loss is more robust in this regime but still produces weaker invariance results than on models with low collision rates. Developing losses or architectural modifications that can handle high-collision-rate feature spaces is left open.

## 5.2 Future Work

There are several promising directions for future work in this area. First, we pose the question: Are there other losses that can be used for post-training augmentation invariance? We have shown experimentally that the MaWa and WaCo losses are suitable candidates, though further theoretical investigation of their properties, especially the WaCo loss, is still needed. Finally, we highlight the experiments on noise invariance, since these results suggest that these methods may be promising for adversarial robustness, more generally. By treating adversarial attacks as augmentations, an adapter trained on either the MaWa or WaCo loss could potentially produce a robust composite network.

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

# A  Background

## A.1  Wasserstein Distance

We first review the definition of the Wasserstein distance in the general setting of complete and separable metric spaces, which we call Polish metric spaces for short. This material should be standard for anyone familiar with Villani's introductory book Villani (2003). We also recall the definition of the sliced Wasserstein distance, which we use for our computational implementation of the Wasserstein correlation maximization objective discussed in Sections 2.2 and 3.3.2.

Formally, we need the following regularity assumption to ensure that the Wasserstein distance (of order $p$) is indeed an actual metric, as opposed to a generalized metric. We state the condition for completeness, though we gloss over these kinds of nuances in practice.

**Definition A.1** *Let $(X, d_X)$ be Polish metric space, let $\Sigma_X$ denote the Borel $\sigma$-algebra on $X$ induced by $d_X$, and let $1 \leq p < \infty$. A Borel probability measure $\mu$ on $(X, \Sigma_X)$ (also called a distribution) has **finite $p$th moment** if*

$$\int_X d_X^p(x_0, x)\mu(dx) < \infty \tag{40}$$

*for some (and hence, by the triangle inequality, any) $x_0 \in X$. We denote the set of all such probability measures by $\mathcal{P}_p(X)$, and we denote the collection of all Borel probability measures by $\mathcal{P}(X)$.*

**Definition A.2** *Let $\mu \in \mathcal{P}_p(X)$, and let $\nu \in \mathcal{P}_p(Y)$. A **coupling** of $\mu$ and $\nu$ is a probability measure $\pi \in \mathcal{P}(X \times Y)$ with first and second marginals equal to $\mu$ and $\nu$, respectively. That is, for all $A \in \Sigma_X$ and all $B \in \Sigma_Y$,*

$$\pi(A \times Y) = \mu(A), \quad and \quad \pi(X \times B) = \nu(B).$$

*The collection of all couplings between $\mu$ and $\nu$ is denoted by $\Pi(\mu, \nu)$. We will also denote the first and second marginals of $\pi$ by $\pi^1$ and $\pi^2$.*

A coupling can be seen as a relaxation of a transport map, which is a measurable map $T : X \rightarrow Y$ satisfying $T_\sharp \mu = \nu$ where $T_\sharp \mu$ is the **pushforward** of $\mu$ defined by

$$(T_\sharp \mu)(B) = \mu(T^{-1}(B)) \tag{41}$$

for all $B \in \Sigma_Y$. Note also that the space of couplings is always non-empty, since, at the least, it always contains the product measure.

**Definition A.3** *Let $\mu \in \mathcal{P}_p(X)$ and $\nu \in \mathcal{P}_p(Y)$. The **product measure**, or tensor product, of $\mu$ and $\nu$ is the joint probability measure $\mu \otimes \nu$ in $\mathcal{P}(X \times Y)$ defined on test functions $\phi \in C_b(X \times Y)$ by*

$$\int_{X \times Y} \phi(x,y)(\mu \otimes \nu)(dx, dy) = \int_Y \left( \int_X \phi(x,y)\mu(dx) \right) \nu(dy). \tag{42}$$

*Alternatively, we can set*

$$(\mu \otimes \nu)(A \times B) = \mu(A)\nu(B) \tag{43}$$

*for $A \times B \in \Sigma_X \times \Sigma_Y$. Since we always assume that $X \times Y$ is standard Borel, the above equality determines $\mu \otimes \nu$ uniquely on all of $\Sigma_{X \times Y}$.*

**Definition A.4** *Let $(X, d_X)$ be a Polish metric space, and let $1 \le p < \infty$. The **Wasserstein metric of order p***

$$W_{p,d_X} : \mathcal{P}_p(X) \times \mathcal{P}_p(X) \to \mathbb{R}_{\ge 0} \tag{44}$$

*is defined by*

$$W_{p,d_X}(\mu, \nu) = \left( \inf_{\pi \in \Pi(\mu,\nu)} \int_{X \times X} d_X^p(x_1, x_2)\pi(dx_1, dx_2) \right)^{1/p}. \tag{45}$$

*We call the pair $(\mathcal{P}_p(X), W_{p,d_X})$ the **Wasserstein space of order p** associated with $(X, d_X)$.*

When the context is clear, we denote the Wasserstein metric by $W_p$ alone. Likewise, we will sometimes denote the so-called ground metric $d_X$ by $d$ alone. Remarkably, the Wasserstein space is itself a Polish metric space whenever $(X, d_X)$ is.

**Proposition A.5** *Let $(X, d_X)$ be a complete and separable metric space. Then, the Wasserstein space $(\mathcal{P}_p(X), W_{p,d_X})$ is also complete and separable.*

**Proof 1** *See Proposition 7.1.5 in Ambrosio et al. (2008).*

Informally, the Wasserstein construction is said to lift any ground metric $d_X$ to a metric $W_{p,d_X}$ on the space of (sufficiently regular) Borel probability measures over $X$. Further, the Wasserstein metric encodes the ground metric, in the sense that $W_{p,d_X}(\delta_x, \delta_y) = d_X(x, y)$, where $\delta_x$ is the usual Dirac mass defined by

$$\delta_x(E) = \begin{cases} 1 & \text{if } x \in E \\ 0 & \text{if } x \notin E \end{cases} \tag{46}$$

for $E \in \Sigma_X$; see Villani (2003) for more details. In passing, we observe that Definition A.4 is still valid with a lower semicontinuous cost function $c$ on $X$ in place of $d_X$. Custom cost functions are indeed part of the appeal of optimal transport-based distances and are worth further study, though in the present work we restrict our attention to metrics.

To define Wasserstein correlation in Subsection 2.2, we need to equip the product space $X \times Y$ with a metric so that joint distributions can be viewed as elements of a Wasserstein space. We choose to work with the product metric for this.

**Definition A.6** *Let $(X, d_X)$ and $(Y, d_Y)$ be Polish metric spaces, and let $1 \le p < \infty$. Then, the **product metric of order p***

$$d_{p,X \times Y} : (X \times Y) \times (X \times Y) \to \mathbb{R}_{\ge 0} \tag{47}$$

*is defined by*

$$d_{p,X \times Y}\big((x_1, y_1), (x_2, y_2)\big) = \left( d_X^p(x_1, x_2) + d_Y^p(y_1, y_2) \right)^{1/p}. \tag{48}$$

*We'll denote this metric by $d_{XY}$, and we assume that $p$ is always chosen to match the order of the Wasserstein metric.*

The Wasserstein space $(\mathcal{P}_p(X \times Y), W_{p,d_{XY}})$ is the usual Wasserstein space where the ground metric is now $d_{XY}$. Further, it is elementary to check that any coupling $\pi$ of $\mu \in \mathcal{P}_p(X)$ and $\nu \in \mathcal{P}_p(Y)$ has finite $p$th moment with respect to $d_{XY}$. That is, $\Pi(\mu, \nu) \subseteq \mathcal{P}_p(X \times Y)$.

### A.1.1 Sliced Wasserstein Distance

Extensive work has gone into finding efficient algorithms for computing Wasserstein distances. Entropic regularization (Cuturi, 2013) and slicing techniques (Bonneel et al., 2015) are two of the most common approaches, and in our experiments, we use the latter.

**Definition A.7** *Let $\omega$ denote uniform distribution on the unit sphere $\mathbb{S}^{d-1} \subseteq \mathbb{R}^d$. Then, the **sliced Wasserstein distance of order** p*

$$SW_p : \mathcal{P}_p(\mathbb{R}^d) \times \mathcal{P}_p(\mathbb{R}^d) \to \mathbb{R}_{\geq 0} \tag{49}$$

*is defined by*

$$SW_p(\mu,\nu) = \left( \int_{\theta:\mathbb{S}^{d-1}} W_p^p\big((P_\theta)_\sharp \mu, (P_\theta)_\sharp \nu\big) \omega(d\theta) \right)^{1/p}, \tag{50}$$

*where $P_\theta$ is the linear form given by $P_\theta(x) = \theta^T x$.*

See Bonnotte (2013) for a proof that the sliced Wasserstein distance is an actual metric. To compute sliced distances, we use the closed-form solution to the Wasserstein distance in the case of 1-dimensional distributions on $\mathbb{R}$. For $\mu \in \mathcal{P}_p(\mathbb{R})$, define the cumulative distribution function (CDF) $F_\mu : \mathbb{R} \to [0,1]$ by $F_\mu(x) = \mu\big((-\infty, x]\big)$, and define the quantile function $F_\mu^{-1} : [0,1] \to \mathbb{R}$ by

$$F_\mu^{-1}(t) = \inf\{x : F_\mu(x) \geq t\}. \tag{51}$$

Then, as shown in Rachev & Rüschendorf (2006),

$$W_p(\mu,\nu) = \left( \int_0^1 \left| F_\mu^{-1}(t) - F_\nu^{-1}(t) \right|^p dt \right)^{1/p}. \tag{52}$$

Letting $F_{\theta,\mu}^{-1}$ denote the quantile function of $(P_\theta)_\sharp \mu$, we have

$$SW_p(\mu,\nu) = \left( \int_{\mathbb{S}^{d-1}} \left( \int_0^1 \left| F_{\theta,\mu}^{-1}(t) - F_{\theta,\nu}^{-1}(t) \right|^p dt \right) \omega(d\theta) \right)^{1/p}. \tag{53}$$

In practice, given empirical distributions, we compute the one-dimensional Wasserstein distance by sorting the points of those distributions and then taking their average Euclidean distance. Our exact computational implementation follows the work of Nguyen et al. (2022).

## A.2 Markov-Wasserstein Kernels

Abstractly, we model encoders as Markov-Wasserstein kernels, which we introduce next. Ordinary Markov kernels are defined as follows.

**Definition A.8** *Let $(X, \Sigma_X)$ and $(Y, \Sigma_Y)$ be measurable spaces. A **Markov kernel** from $(X, \Sigma_X)$ to $(Y, \Sigma_Y)$ is a map $F : \Sigma_Y \times X \to [0,1]$ satisfying*

    *1. $F(-|x) : \Sigma_Y \to [0,1]$ is a probability measure for any fixed $x \in X$.*

    *2. The map $x \mapsto F(E|x)$ is $\Sigma_X$-measurable for any fixed $E \in \Sigma_Y$.*

*Note that the conditional notation $F(-|x)$ above is synonymous with $F(-,x)$.*

Equivalently, a Markov kernel can be viewed as a measurable map $F : X \to \mathcal{P}(Y)$, where $\mathcal{P}(Y)$ is equipped with the initial $\sigma$-algebra with respect to all evaluation maps $\text{ev}_E : \mathcal{P}(Y) \to [0,1]$, $\text{ev}_E(\mu) = \mu(E)$, for $E \in \Sigma_Y$; see Lemma 3.1 in Kallenberg (2021).

Both of these definitions are purely measure-theoretic, whereas, for our purposes, we would like to work in a metric-enhanced setting. To do this, we impose in Definition A.8 the additional constraint that each distribution $F(-|x)$ should have finite $p$th moment. Then, we additionally impose the following regularity condition to equip the resulting collection of Markov kernels with a metric.

**Definition A.9** *Let $(X, d_X, \mu_X)$ be a metric measure space (i.e., a Polish metric space equipped with a probability measure $\mu_X$ on the Borel $\sigma$-algebra induced by $d_X$), and let $(Y, d_Y)$ be a Polish metric space. A Markov kernel $F : X \to \mathcal{P}_p(Y)$ has **finite pth moment** if*

$$\int_X \left( \int_Y d_Y^p(y_0, y) F(dy|x) \right) \mu_X(dx) < \infty \tag{54}$$

*for some (and hence any) $y_0 \in Y$. We let $\mathcal{K}_{\mu_X}^p(X, Y)$ denote the collection of all such Markov kernels.*

In Patterson (2021), Patterson shows how to equip $\mathcal{K}_{\mu_X}^p(X, Y)$ with a metric that closely resembles the ordinary Wasserstein metric using a generalized notion of coupling between two Markov kernels, and he additionally shows (in Proposition 5.5) that the metric can be computed as a generalized $L^p$ metric.

**Definition A.10** *Let $(X, d_X, \mu_X)$ be a metric measure space, and let $(Y, d_Y)$ be a Polish metric space. The $\mathbf{L^p}$ **space** $L_{\mu_X}^p(X, Y)$ is the set of $\mu_X$-a.e. equal equivalence classes of measurable functions $f : X \to Y$ satisfying*

$$\int_X d_Y^p \big(y_0, f(x)\big) \mu_X(dx) < \infty \tag{55}$$

*for some (and hence any) $y_0 \in Y$. We say that $f$ is $\mathbf{L^p}$ **integrable** if this condition is met. Further, $L_{\mu_X}^p(X, Y)$ is a metric space when equipped with the $\mathbf{L^p}$ **metric***

$$d_{L^p} : L_{\mu_X}^p(X, Y) \times L_{\mu_X}^p(X, Y) \to \mathbb{R}_{\geq 0} \tag{56}$$

*defined by*

$$d_{L^p}(f, g) = \left( \int_X d_Y^p \big(f(x), g(x)\big) \mu_X(dx) \right)^{1/p} \tag{57}$$

*for $1 \leq p < \infty$.*

For the case of $\mathcal{K}_{\mu_X}^p(X, Y)$, in particular, $d_Y$ in the above is taken to be the Wasserstein metric $W_{p, d_Y}$. This discussion so far applies to Markov kernels as defined in Definition A.8 with the extra regularity conditions noted before. Similar to the purely measure-theoretic case, though, we can equivalently view a Markov kernel $F$ taking values in a Wasserstein space $\mathcal{P}_p(Y)$ as a measurable map $F : X \to \mathcal{P}_p(Y)$, where $\mathcal{P}_p(Y)$ is equipped with the Borel $\sigma$-algebra induced by the Wasserstein metric; see Eikenberry (2023) for details. This will be the perspective we adopt in the present work.

**Remark A.11** *That is, we work with the generalized $L^p$ space $L_{\mu_X}^p(X, \mathcal{P}_p(Y))$ of Wasserstein-valued Markov kernels $F : X \to \mathcal{P}_p(Y)$, which we call the space of **Markov-Wasserstein kernels**. Further, we will denote the generalized $L^p$ metric from Definition A.10 by $MW_p$, and we call it the **Markov-Wasserstein metric**. We define the Markov-Wasserstein minimization loss in Section 3.3.1. Finally, we note that the two approaches to defining Markov-Wasserstein kernels presented here lead to isometrically isomorphic metric spaces; see Theorem 2.25 in Eikenberry (2023).*

Markov-Wasserstein kernels (and also general Markov kernels) are closely connected to joint distributions via disintegrations. The remaining definitions and theorems in this subsection are usually only defined for the measure-theoretic case, but, in fact, everything carries over seamlessly to the metric-enhanced setting.

**Definition A.12** *Let $\gamma \in \mathcal{P}_p(X \times Y)$ with marginals $\gamma^1 = \mu$ and $\gamma^2 = \nu$. Then, a Markov-Wasserstein kernel $F \in L_\mu^p(X, \mathcal{P}_p(Y))$ is a called a **disintegration** of $\gamma$ if for all $\phi \in C_b(X \times Y)$,*

$$\int_{X \times Y} \phi(x, y) \gamma(dx, dy) = \int_X \left( \int_Y \phi(x, y) F_x(dy) \right) \mu(dx). \tag{58}$$

In Villani (2003), Villani writes $\int_X \big(\delta_x \otimes F_x\big) \mu(dx)$ for the joint distribution defined by the iterated integral above. As shown in Eikenberry (2023), it is also possible to define this measure as a literal Lebesgue-type

integral for Markov-Wasserstein kernels. We will not review the details, since we primarily use this machinery to give a precise mathematical definition for the Wasserstein correlation maximization objective.

The well-known Disintegration Theorem states that disintegrations of (sufficiently nice) joint distributions always exist.

**Theorem A.13 (Lᵖ Disintegration Theorem)** *Let $\gamma \in (\mathcal{P}_p(X \times Y), W_{p,d_{XY}})$ with $\gamma^1 = \mu$ and $\gamma^2 = \nu$. Then, there exist almost surely unique Markov-Wasserstein kernels $F \in L^p_\mu(X, \mathcal{P}_p(Y))$ and $G \in L^p_\nu(Y, \mathcal{P}_p(X))$ with*

$$\int_X \big(\delta_x \otimes F_x\big)\mu(dx) = \gamma = \int_Y \big(G_y \otimes \delta_y\big)\nu(dy). \tag{59}$$

**Proof 2** *See Theorem 10.4.5 of Bogachev & Ruas (2007) for the classic Disintegration Theorem. Also, see Theorem 3.16 in Eikenberry (2023) for the extension to the $L^p$ case and for an interpretation of this result as an equality of actual integrals.*

We can use disintegrations to give a precise definition of Bayesian inverse maps. Markov-kernel formulations of Bayesian inference are well-developed in probabilistic programming semantics and in category-theoretic approaches to probability; see, for example, Cho & Jacobs (2019); Clerc et al. (2017); Culbertson & Sturtz (2014); Fritz (2020). We again state things for the metric-enhanced case. First, we need to define how to push a measure forward through a Markov, or Markov-Wasserstein, kernel, rather than a deterministic map.

**Definition A.14** *Let $F : X \to \mathcal{P}_p(Y)$ be a Markov-Wasserstein kernel, and let $\mu \in \mathcal{P}_p(X)$. The **generalized pushforward** of $\mu$ under $F$, or, alternatively, the **Kleisli composition**[8] of $F$ and $\mu$, is the probability measure $F \odot \mu$ in $\mathcal{P}_p(Y)$ defined by*

$$(F \odot \mu)(B) = \int_X F(B|x)\mu(dx) \tag{60}$$

*for $B \in \Sigma_Y$. Note that if $\gamma = \int_X \big(\delta_x \otimes F_x\big)\mu(dx)$, then $\gamma^2 = F \odot \mu$, and we call this the **marginal likelihood** in Bayesian contexts.*

**Definition A.15** *Let $F \in L^p_\mu(X, \mathcal{P}_p(Y))$, and set $\nu := F \odot \mu$. Then, the **Bayesian inverse** $F^\dagger_\mu$ of $F$ with respect to $\mu$ is the $\nu$-almost surely unique Markov-Wasserstein kernel in $L^p_\nu(Y, \mathcal{P}_p(X))$ satisfying*

$$\int_X \big(\delta_x \otimes F(x)\big)\mu(dx) = \int_Y \big(F^\dagger_\mu(y) \otimes \delta_y\big)(F \odot \mu)(dy). \tag{61}$$

*Note that $F^\dagger_\mu$ is guaranteed to exist by Theorem A.13.*

Finally, we submit the following as a highly general definition of an autoencoder that captures the statistical essence of many models without any secondary constraints, such as smoothness, determinism, or stricter inversion requirements.

**Definition A.16** *Let $\mu_X \in \mathcal{P}_p(X)$ be a (data) distribution. Then, a **probabilistic autoencoder**, or simply an **autoencoder**, for $\mu_X$ is a pair $(E, D)$ of Markov-Wasserstein kernels $E : X \to \mathcal{P}_p(Z)$ and $D : Z \to \mathcal{P}_p(X)$ such that $D$ is a Bayesian inverse of $E$ with respect to $\mu_X$. That is,*

$$\int_X \big(\delta_x \otimes E(x)\big)\mu_X(dx) = \int_Z \big(D(z) \otimes \delta_z\big)(E \odot \mu_X)(dz). \tag{62}$$

*(We switch to $Z$ here instead of $Y$ to match the convention for latent spaces.)*

---

[8]The Kleisli terminology comes from the fact that this operation can be viewed as an instance of composition in a Kleisli category. That is, it can be seen as an ordinary pushforward operation into an iterated probability space (i.e., a space of distributions of distributions), followed by an averaging operation. Details on Kleisli categories can be found in Fritz (2020), among other sources, but we will only need the more straightforward definition given here.

In practice, we work with deterministic (auto)encoders (specifically simple MLPs for our tests), which can still be seen as Markov-Wasserstein kernels (or simply Markov kernels) via Dirac embeddings. That is, given a measurable function $f : X \to Z$, we obtain a Markov kernel $F : X \to \mathcal{P}(Z)$ by setting $F(x) = \delta_{f(x)}$. Further, we can place additional integrability or smoothness constraints on $f$, which, in turn, induce constraints on the corresponding kernel $F$. In the main text, we show that even when starting with a deterministic encoder, data augmentation naturally leads to more general, probabilistic encoders as the main object of study.

## B   Experiment Details

We use simple MLPs with ReLU activations for all tests. For the adapter networks $E_\theta$, the dimensions are of the form $(\mathrm{input}, \mathrm{hidden}, \mathrm{output}) = (d, 4096, d)$, where $d$ is the dimension of the pretrained latent space, namely $d = 384$ for DINO (transformer backbone), $d = 512$ for CLIP, or $d = 2048$ for SwAV, R-DINO (DINO with ResNet backbone), and ResNet50. For dimensionality reduction with WaCo, the networks have dimensions $(d, 4096, d/4)$: concretely, $(384, 4096, 96)$ for DINO, $(512, 4096, 128)$ for CLIP, and $(2048, 4096, 512)$ for SwAV, R-DINO, and ResNet50. Our final MNIST test uses a one-hidden-layer network with dimensions $(784, 4096, 64)$.

For classifiers, we use three configurations: a linear classifier of dimensions $(d, k)$, a nonlinear classifier with one hidden layer of dimensions $(d, 4096, k)$, and an end-to-end classifier of dimensions $(d, 4096, 4096, k)$, where $k$ is the number of classes. When classifying from a WaCo-reduced latent space of dimension $d/4$, the linear and nonlinear classifiers start from $d/4$ instead of $d$.

In all of our tests, at least for the MaWa and WaCo losses, we found comparable results across a wide range of choices for the batch size, number of epochs, learning rate, optimizer, scheduler, and number of augmentation samples. Default settings are listed in Table 12. The one exception is that for SimCLR, we set the batch size to 1024 to match the total number of samples used in the other three losses, namely, $3 \times 256$ augmented samples plus an additional 256 non-augmented samples for a total of 1024.

All models were trained on either a single NVIDIA RTX 3070 GPU, a NVIDIA RTX 3090 GPU, or a NVIDIA A100 GPU. Compute times are fairly modest (on the order of tens of minutes), but when training for invariance to complicated, composite augmentations, training can extend to multiple hours when using only a single GPU. Applying multiple augmentations and extracting pretrained features are two of the main bottlenecks.

Table 12: Default hyperparameters

| Parameter | Value |
| --- | --- |
| Batch size | 256 |
| Epochs | 100 |
| Learning rate | $1 \times 10^{-3}$ |
| Optimizer | AdamW with weight decay $1 \times 10^{-4}$ |
| Scheduler | CosineAnnealing with minimum rate $4 \times 10^{-4}$ |
| Augmentation Samples $s$ | 3 |

For our invariance tests, all classifiers are trained on a standard cross-entropy loss, and the optimizer and scheduler settings are the same as in Table 12 but without weight decay for the optimizer (meaning that we are effectively using an ordinary Adam optimizer rather than AdamW (Loshchilov & Hutter, 2017) for classification tests).

We use the Kornia package (Riba et al., 2020) for augmentations. We vary the augmentation parameters for each element of the batch (same_on_batch = False), but, in fact, results are comparable when the same augmentation is applied across the batch, which leads to better computational efficiency, when needed.

Default augmentation parameters are listed in Table 13. We take, for example, 50 to 70 percent crops with aspect ratio $(0.75, 1.33)$ and then resize to the original dimensions of the data.

Table 13: Default augmentation parameters

| Augmentation | Parameters | |
|---|---|---|
| RandomRotation | degrees $= (-180, 180)$ | |
| RandomAffine | degrees $= (-30, 30)$, | translate $= (0.2, 0.2)$, |
| | scale $= (0.8, 1.2)$, | shear $= (-15, 15)$ |
| RandomGaussianNoise | mean $= 0$, | std $= 1$ |
| RandomResizedCrop | scale $= (0.5, 0.7)$, | resize to $= (0.75, 1.33)$ |

## C  Additional Experiments

We report here the full suite of structure-preservation and classification results for all pretrained networks and augmentation types not included in the main text. Specifically, we provide:

- Structure-preservation tables for DINO on affine and crop augmentations and SimCLR/HSIC comparisons (Tables 14–17).
- Remaining results for SwAV, including affine/crop tables for MaWa and WaCo losses as well as full SimCLR/HSIC comparisons (Tables 18–23).
- Results for R-DINO (DINO with ResNet backbone, $d = 2048$) on all augmentation types (Tables 24–27).
- Results for CLIP ($d = 512$) on all augmentation types (Tables 28–31).
- Results for supervised ResNet50 ($d = 2048$) on all augmentation types (Tables 32–35).
- Results on TinyImageNet (200 classes) and a 50-class TinyImageNet subset (Tables 36–43) for DINO. The subset was selected via a greedy procedure to maximize pairwise class separation in the DINO latent space.
- Collision rate tables for all models not reported in the main text (Tables 44–47).

The overall patterns are consistent with those reported in the main text: The MaWa loss achieves the strongest combination of invariance and structure preservation across all self-supervised models, the WaCo loss provides a competitive alternative with the additional capability of dimensionality reduction, and the SimCLR and HSIC losses consistently corrupt the pretrained latent space. The supervised ResNet50 is the one setting where MaWa fails, as predicted by its high residual collision rates (see Table 11 in the main text).

Table 14: MaWa and WaCo affine-invariant encoders on DINO features.

| Structure | | STL10 + DINO + Affine | | |
|---|---|---|---|---|
| Preservation | | MaWa ($d = 384$) | WaCo ($d = 384$) | WaCo ($d = 96$) |
| Similarity | $L_1 \mid L_2$ | 0.82 │ 1.07 | 0.46 │ 1.66 | 0.46 │ 1.36 |
| Tests | $m \mid b$ | 1.00 │ -0.61 | 1.16 │ -20.56 | 1.19 │ -27.25 |
| | $R^2$ | 0.98 | 0.63 | 0.74 |
| | RMSD | 1.37 | 7.90 | 8.96 |
| | NRMSD | 0.01 | 0.08 | 0.09 |
| | CVRMSD | 0.01 | 0.07 | 0.08 |
| Classification | LC | 98.19 │ 97.20 | 98.19 │ 96.33 | 97.51 │ 95.97 |
| | NC | 98.65 │ 97.43 | 98.42 │ 96.83 | 98.21 │ 96.85 |
| | EC | 98.58 │ 94.65 | 98.58 │ 94.65 | 98.58 │ 94.65 |

Table 15: MaWa and WaCo crop-invariant encoders on DINO features.

| Structure | | STL10 + DINO + Crop | | |
|---|---|---|---|---|
| Preservation | | MaWa ($d = 384$) | WaCo ($d = 384$) | WaCo ($d = 96$) |
| Similarity | $L_1 \mid L_2$ | 0.80 │ 1.04 | 0.49 │ 1.85 | 0.52 │ 1.55 |
| Tests | $m \mid b$ | 1.00 │ -2.10 | 1.20 │ -20.86 | 1.18 │ -25.41 |
| | $R^2$ | 0.95 | 0.62 | 0.76 |
| | RMSD | 2.57 | 8.12 | 7.66 |
| | NRMSD | 0.02 | 0.08 | 0.07 |
| | CVRMSD | 0.02 | 0.07 | 0.07 |
| Classification | LC | 98.20 │ 98.11 | 97.97 │ 97.66 | 97.65 │ 97.14 |
| | NC | 98.66 │ 98.36 | 98.32 │ 98.00 | 98.50 │ 98.12 |
| | EC | 98.58 │ 98.10 | 98.58 │ 98.10 | 98.58 │ 98.10 |

Table 16: SimCLR and HSIC affine-invariant encoders on DINO features.

| Structure | | STL10 + DINO + Affine | |
|---|---|---|---|
| Preservation | | SCLR ($d = 384$) | HSIC ($d = 384$) |
| Similarity | $L_1 \mid L_2$ | 0.12 │ 13.95 | 0.20 │ 8.28 |
| Tests | $m \mid b$ | 4.32 │ -189.90 | 2.27 │ -83.79 |
| | $R^2$ | 0.06 | 0.03 |
| | RMSD | 231.68 | 128.49 |
| | NRMSD | 2.24 | 1.24 |
| | CVRMSD | 2.08 | 1.15 |
| Classification | LC | 97.40 │ 95.90 | 96.73 │ 90.64 |
| | NC | 97.41 │ 95.71 | 97.24 │ 92.10 |
| | EC | 98.58 │ 94.65 | 98.58 │ 94.65 |

Table 17: SimCLR and HSIC crop-invariant encoders on DINO features.

| Structure | | STL10 + DINO + Crop | |
|---|---|---|---|
| Preservation | | SCLR ($d = 384$) | HSIC ($d = 384$) |
| Similarity | $L_1 \mid L_2$ | 0.03 \| 6.72 | 0.25 \| 11.87 |
| Tests | $m \mid b$ | 2.71 \| -146.27 | 3.70 \| -106.78 |
| | $R^2$ | 0.06 | 0.03 |
| | RMSD | 104.04 | 255.11 |
| | NRMSD | 1.01 | 2.47 |
| | CVRMSD | 0.93 | 2.29 |
| Classification | LC | 97.85 \| 97.60 | 95.06 \| 93.93 |
| | NC | 97.92 \| 97.64 | 96.56 \| 95.76 |
| | EC | 98.58 \| 98.10 | 98.58 \| 98.10 |

Table 18: MaWa and WaCo affine-invariant encoders on SwAV features.

| Structure | | STL10 + SwAV + Affine | | |
|---|---|---|---|---|
| Preservation | | MaWa ($d = 2048$) | WaCo ($d = 2048$) | WaCo ($d = 512$) |
| Similarity | $L_1 \mid L_2$ | 0.46 \| 0.94 | 0.39 \| 1.85 | 0.42 \| 1.47 |
| Tests | $m \mid b$ | 0.95 \| -1.71 | 1.69 \| -6.70 | 1.26 \| -2.90 |
| | $R^2$ | 0.90 | 0.82 | 0.82 |
| | RMSD | 2.32 | 2.10 | 1.02 |
| | NRMSD | 0.13 | 0.12 | 0.06 |
| | CVRMSD | 0.20 | 0.18 | 0.09 |
| Classification | LC | 93.70 \| 88.42 | 90.09 \| 86.00 | 91.39 \| 87.70 |
| | NC | 93.46 \| 87.74 | 90.26 \| 86.05 | 91.38 \| 87.64 |
| | EC | 94.23 \| 76.06 | 94.23 \| 76.06 | 94.23 \| 76.06 |

Table 19: MaWa and WaCo crop-invariant encoders on SwAV features.

| Structure | | STL10 + SwAV + Crop | | |
|---|---|---|---|---|
| Preservation | | MaWa ($d = 2048$) | WaCo ($d = 2048$) | WaCo ($d = 512$) |
| Similarity | $L_1 \mid L_2$ | 0.41 \| 0.93 | 0.32 \| 1.78 | 0.36 \| 1.47 |
| Tests | $m \mid b$ | 0.91 \| -1.65 | 1.72 \| -7.00 | 1.38 \| -4.25 |
| | $R^2$ | 0.87 | 0.83 | 0.86 |
| | RMSD | 2.77 | 2.10 | 1.07 |
| | NRMSD | 0.15 | 0.12 | 0.06 |
| | CVRMSD | 0.24 | 0.18 | 0.09 |
| Classification | LC | 93.73 \| 94.32 | 90.74 \| 91.39 | 92.07 \| 92.52 |
| | NC | 93.58 \| 93.98 | 91.14 \| 91.63 | 92.12 \| 92.34 |
| | EC | 94.23 \| 94.33 | 94.23 \| 94.33 | 94.23 \| 94.33 |

Table 20: SimCLR and HSIC rotation-invariant encoders on SwAV features.

| Structure | | STL10 + SwAV + Rotation | |
| --- | --- | --- | --- |
| Preservation | | SCLR ($d = 2048$) | HSIC ($d = 2048$) |
| Similarity | $L_1 \mid L_2$ | 0.25 \| 15.98 | 0.07 \| 0.83 |
| Tests | $m \mid b$ | 3.80 \| -0.56 | 0.09 \| 2.38 |
| | $R^2$ | 0.08 | 0.04 |
| | RMSD | 37.92 | 8.29 |
| | NRMSD | 2.11 | 0.46 |
| | CVRMSD | 3.28 | 0.72 |
| Classification | LC | 79.50 \| 76.31 | 71.90 \| 69.49 |
| | NC | 79.69 \| 76.78 | 81.67 \| 77.83 |
| | EC | 94.23 \| 50.35 | 94.23 \| 50.35 |

Table 21: SimCLR and HSIC noise-invariant encoders on SwAV features.

| Structure | | STL10 + SwAV + Noise | |
| --- | --- | --- | --- |
| Preservation | | SCLR ($d = 2048$) | HSIC ($d = 2048$) |
| Similarity | $L_1 \mid L_2$ | 0.11 \| 9.56 | 0.06 \| 0.86 |
| Tests | $m \mid b$ | 1.28 \| 10.79 | 0.04 \| 3.12 |
| | $R^2$ | 0.02 | 0.00 |
| | RMSD | 20.18 | 8.19 |
| | NRMSD | 1.12 | 0.46 |
| | CVRMSD | 1.75 | 0.71 |
| Classification | LC | 73.94 \| 46.88 | 65.83 \| 59.97 |
| | NC | 76.39 \| 47.06 | 74.38 \| 65.05 |
| | EC | 94.23 \| 40.82 | 94.23 \| 40.82 |

Table 22: SimCLR and HSIC affine-invariant encoders on SwAV features.

| Structure | | STL10 + SwAV + Affine | |
| --- | --- | --- | --- |
| Preservation | | SCLR ($d = 2048$) | HSIC ($d = 2048$) |
| Similarity | $L_1 \mid L_2$ | 0.39 \| 15.23 | 0.07 \| 0.70 |
| Tests | $m \mid b$ | 5.08 \| -8.27 | 0.10 \| 2.06 |
| | $R^2$ | 0.15 | 0.07 |
| | RMSD | 44.08 | 8.52 |
| | NRMSD | 2.45 | 0.47 |
| | CVRMSD | 3.82 | 0.74 |
| Classification | LC | 87.56 \| 85.62 | 81.24 \| 78.98 |
| | NC | 87.78 \| 86.06 | 87.96 \| 85.62 |
| | EC | 94.23 \| 76.06 | 94.23 \| 76.06 |

Table 23: SimCLR and HSIC crop-invariant encoders on SwAV features.

| Structure | | STL10 + SwAV + Crop | |
|---|---|---|---|
| Preservation | | SCLR ($d = 2048$) | HSIC ($d = 2048$) |
| Similarity | $L_1 \mid L_2$ | 0.12 \| 14.15 | 0.07 \| 0.77 |
| Tests | $m \mid b$ | 5.80 \| -26.67 | 0.10 \| 2.01 |
| | $R^2$ | 0.21 | 0.09 |
| | RMSD | 34.84 | 8.55 |
| | NRMSD | 1.94 | 0.48 |
| | CVRMSD | 3.02 | 0.74 |
| Classification | LC | 92.92 \| 93.34 | 87.95 \| 87.94 |
| | NC | 92.33 \| 92.84 | 92.67 \| 92.69 |
| | EC | 94.23 \| 94.33 | 94.23 \| 94.33 |

Table 24: MaWa and WaCo rotation-invariant encoders on R-DINO features.

| Structure | | STL10 + R-DINO + Rotation | | |
|---|---|---|---|---|
| Preservation | | MaWa ($d = 2048$) | WaCo ($d = 2048$) | WaCo ($d = 512$) |
| Similarity | $L_1 \mid L_2$ | 0.47 \| 0.96 | 0.35 \| 2.42 | 0.45 \| 1.73 |
| Tests | $m \mid b$ | 0.92 \| -1.32 | 1.84 \| -8.31 | 1.47 \| -4.76 |
| | $R^2$ | 0.95 | 0.88 | 0.89 |
| | RMSD | 2.29 | 2.88 | 1.72 |
| | NRMSD | 0.11 | 0.14 | 0.09 |
| | CVRMSD | 0.20 | 0.25 | 0.15 |
| Classification | LC | 94.25 \| 82.71 | 85.89 \| 77.72 | 88.75 \| 81.38 |
| | NC | 93.77 \| 82.04 | 86.59 \| 78.40 | 89.50 \| 81.66 |
| | EC | 94.76 \| 53.11 | 94.76 \| 53.11 | 94.76 \| 53.11 |

Table 25: MaWa and WaCo affine-invariant encoders on R-DINO features.

| Structure | | STL10 + R-DINO + Affine | | |
|---|---|---|---|---|
| Preservation | | MaWa ($d = 2048$) | WaCo ($d = 2048$) | WaCo ($d = 512$) |
| Similarity | $L_1 \mid L_2$ | 0.50 \| 0.96 | 0.26 \| 2.58 | 0.38 \| 2.01 |
| Tests | $m \mid b$ | 0.93 \| -1.29 | 1.81 \| -8.06 | 1.63 \| -6.05 |
| | $R^2$ | 0.95 | 0.83 | 0.86 |
| | RMSD | 2.14 | 2.97 | 2.47 |
| | NRMSD | 0.11 | 0.15 | 0.12 |
| | CVRMSD | 0.18 | 0.26 | 0.21 |
| Classification | LC | 94.41 \| 89.00 | 88.61 \| 84.39 | 91.50 \| 87.17 |
| | NC | 93.97 \| 88.27 | 89.18 \| 84.69 | 91.55 \| 86.82 |
| | EC | 94.76 \| 79.27 | 94.76 \| 79.27 | 94.76 \| 79.27 |

Table 26: MaWa and WaCo noise-invariant encoders on R-DINO features.

| Structure | | STL10 + R-DINO + Noise | | |
|---|---|---|---|---|
| Preservation | | MaWa ($d = 2048$) | WaCo ($d = 2048$) | WaCo ($d = 512$) |
| Similarity | $L_1 \mid L_2$ | 0.58 \| 0.96 | 0.28 \| 2.57 | 0.43 \| 1.64 |
| Tests | $m \mid b$ | 0.93 \| -1.02 | 2.04 \| -10.03 | 1.21 \| -2.42 |
| | $R^2$ | 0.96 | 0.84 | 0.86 |
| | RMSD | 1.90 | 3.74 | 1.23 |
| | NRMSD | 0.09 | 0.19 | 0.06 |
| | CVRMSD | 0.16 | 0.32 | 0.11 |
| Classification | LC | 94.51 \| 75.12 | 86.17 \| 68.03 | 88.61 \| 71.00 |
| | NC | 94.12 \| 75.07 | 86.78 \| 67.94 | 88.78 \| 70.69 |
| | EC | 94.76 \| 35.97 | 94.76 \| 35.97 | 94.76 \| 35.97 |

Table 27: MaWa and WaCo crop-invariant encoders on R-DINO features.

| Structure | | STL10 + R-DINO + Crop | | |
|---|---|---|---|---|
| Preservation | | MaWa ($d = 2048$) | WaCo ($d = 2048$) | WaCo ($d = 512$) |
| Similarity | $L_1 \mid L_2$ | 0.36 \| 0.92 | 0.28 \| 2.24 | 0.36 \| 1.96 |
| Tests | $m \mid b$ | 0.83 \| -1.34 | 1.92 \| -9.40 | 1.67 \| -6.88 |
| | $R^2$ | 0.89 | 0.87 | 0.88 |
| | RMSD | 3.40 | 3.01 | 2.27 |
| | NRMSD | 0.17 | 0.15 | 0.11 |
| | CVRMSD | 0.29 | 0.26 | 0.19 |
| Classification | LC | 94.00 \| 93.99 | 89.91 \| 89.78 | 91.19 \| 91.25 |
| | NC | 94.01 \| 93.77 | 90.12 \| 89.89 | 91.24 \| 91.06 |
| | EC | 94.76 \| 94.37 | 94.76 \| 94.37 | 94.76 \| 94.37 |

Table 28: MaWa and WaCo rotation-invariant encoders on CLIP features.

| Structure | | STL10 + CLIP + Rotation | | |
|---|---|---|---|---|
| Preservation | | MaWa ($d = 512$) | WaCo ($d = 512$) | WaCo ($d = 128$) |
| Similarity | $L_1 \mid L_2$ | 0.71 \| 1.18 | 0.54 \| 1.43 | 0.52 \| 1.35 |
| Tests | $m \mid b$ | 0.99 \| -0.01 | 1.12 \| -0.90 | 1.10 \| -0.85 |
| | $R^2$ | 0.99 | 0.80 | 0.81 |
| | RMSD | 0.13 | 0.63 | 0.57 |
| | NRMSD | 0.01 | 0.06 | 0.06 |
| | CVRMSD | 0.01 | 0.07 | 0.06 |
| Classification | LC | 99.08 \| 94.82 | 98.96 \| 92.60 | 99.00 \| 92.93 |
| | NC | 99.06 \| 94.71 | 98.91 \| 91.50 | 99.06 \| 92.62 |
| | EC | 99.09 \| 88.16 | 99.09 \| 88.16 | 99.09 \| 88.16 |

Table 29: MaWa and WaCo affine-invariant encoders on CLIP features.

| Structure | | STL10 + CLIP + Affine | | |
| --- | --- | --- | --- | --- |
| Preservation | | MaWa ($d = 512$) | WaCo ($d = 512$) | WaCo ($d = 128$) |
| Similarity | $L_1 \mid L_2$ | 0.76 \| 1.17 | 0.53 \| 1.34 | 0.52 \| 1.27 |
| Tests | $m \mid b$ | 1.00 \| -0.04 | 1.17 \| -1.20 | 1.10 \| -0.79 |
| | $R^2$ | 0.99 | 0.86 | 0.85 |
| | RMSD | 0.11 | 0.63 | 0.50 |
| | NRMSD | 0.01 | 0.06 | 0.05 |
| | CVRMSD | 0.01 | 0.07 | 0.06 |
| Classification | LC | 99.09 \| 97.59 | 99.10 \| 97.14 | 99.02 \| 97.16 |
| | NC | 99.02 \| 97.57 | 98.99 \| 96.94 | 99.12 \| 97.08 |
| | EC | 99.09 \| 97.07 | 99.09 \| 97.07 | 99.09 \| 97.07 |

Table 30: MaWa and WaCo noise-invariant encoders on CLIP features.

| Structure | | STL10 + CLIP + Noise | | |
| --- | --- | --- | --- | --- |
| Preservation | | MaWa ($d = 512$) | WaCo ($d = 512$) | WaCo ($d = 128$) |
| Similarity | $L_1 \mid L_2$ | 0.92 \| 1.07 | 0.38 \| 3.94 | 0.57 \| 1.60 |
| Tests | $m \mid b$ | 1.00 \| 0.00 | 2.75 \| -10.87 | 1.27 \| -2.12 |
| | $R^2$ | 1.00 | 0.31 | 0.80 |
| | RMSD | 0.05 | 6.59 | 0.77 |
| | NRMSD | 0.00 | 0.65 | 0.08 |
| | CVRMSD | 0.01 | 0.74 | 0.09 |
| Classification | LC | 99.09 \| 78.35 | 98.54 \| 72.11 | 98.81 \| 74.61 |
| | NC | 99.02 \| 78.29 | 98.49 \| 71.47 | 98.89 \| 73.40 |
| | EC | 99.09 \| 54.36 | 99.09 \| 54.36 | 99.09 \| 54.36 |

Table 31: MaWa and WaCo crop-invariant encoders on CLIP features.

| Structure | | STL10 + CLIP + Crop | | |
| --- | --- | --- | --- | --- |
| Preservation | | MaWa ($d = 512$) | WaCo ($d = 512$) | WaCo ($d = 128$) |
| Similarity | $L_1 \mid L_2$ | 0.73 \| 1.07 | 0.53 \| 1.35 | 0.55 \| 1.18 |
| Tests | $m \mid b$ | 1.00 \| -0.04 | 1.17 \| -1.36 | 1.06 \| -0.71 |
| | $R^2$ | 0.99 | 0.88 | 0.88 |
| | RMSD | 0.13 | 0.52 | 0.45 |
| | NRMSD | 0.01 | 0.05 | 0.04 |
| | CVRMSD | 0.02 | 0.06 | 0.05 |
| Classification | LC | 99.14 \| 97.80 | 99.11 \| 97.25 | 99.05 \| 97.28 |
| | NC | 99.06 \| 97.74 | 99.08 \| 97.09 | 99.02 \| 97.28 |
| | EC | 99.09 \| 97.21 | 99.09 \| 97.21 | 99.09 \| 97.21 |

Table 32: MaWa and WaCo rotation-invariant encoders on supervised ResNet50 features.

| Structure | | STL10 + ResNet50 + Rotation | | |
|---|---|---|---|---|
| Preservation | | MaWa ($d = 2048$) | WaCo ($d = 2048$) | WaCo ($d = 512$) |
| Similarity | $L_1 \mid L_2$ | 0.00 \| 0.77 | 0.43 \| 1.80 | 0.52 \| 1.47 |
| Tests | $m \mid b$ | 0.50 \| -12.51 | 1.32 \| -13.68 | 0.98 \| -0.68 |
| | $R^2$ | 0.24 | 0.80 | 0.82 |
| | RMSD | 48.24 | 11.59 | 5.68 |
| | NRMSD | 0.47 | 0.11 | 0.06 |
| | CVRMSD | 0.71 | 0.17 | 0.08 |
| Classification | LC | 50.90 \| 26.21 | 89.14 \| 70.06 | 90.18 \| 71.63 |
| | NC | 50.55 \| 25.38 | 89.99 \| 72.93 | 91.47 \| 74.11 |
| | EC | 92.76 \| 48.63 | 92.76 \| 48.63 | 92.76 \| 48.63 |

Table 33: MaWa and WaCo affine-invariant encoders on supervised ResNet50 features.

| Structure | | STL10 + ResNet50 + Affine | | |
|---|---|---|---|---|
| Preservation | | MaWa ($d = 2048$) | WaCo ($d = 2048$) | WaCo ($d = 512$) |
| Similarity | $L_1 \mid L_2$ | 0.00 \| 0.77 | 0.49 \| 1.76 | 0.55 \| 1.44 |
| Tests | $m \mid b$ | 0.50 \| -12.63 | 1.45 \| -23.27 | 1.10 \| -6.82 |
| | $R^2$ | 0.24 | 0.84 | 0.89 |
| | RMSD | 48.24 | 11.66 | 4.58 |
| | NRMSD | 0.47 | 0.11 | 0.04 |
| | CVRMSD | 0.71 | 0.17 | 0.07 |
| Classification | LC | 50.82 \| 37.15 | 89.99 \| 79.72 | 91.62 \| 81.70 |
| | NC | 50.92 \| 37.67 | 91.39 \| 82.31 | 92.31 \| 83.62 |
| | EC | 92.76 \| 72.80 | 92.76 \| 72.80 | 92.76 \| 72.80 |

Table 34: MaWa and WaCo noise-invariant encoders on supervised ResNet50 features.

| Structure | | STL10 + ResNet50 + Noise | | |
|---|---|---|---|---|
| Preservation | | MaWa ($d = 2048$) | WaCo ($d = 2048$) | WaCo ($d = 512$) |
| Similarity | $L_1 \mid L_2$ | 0.00 \| 0.76 | 0.29 \| 2.14 | 0.49 \| 1.53 |
| Tests | $m \mid b$ | 0.50 \| -12.81 | 1.74 \| -38.74 | 1.01 \| -3.68 |
| | $R^2$ | 0.24 | 0.81 | 0.85 |
| | RMSD | 48.20 | 17.14 | 5.69 |
| | NRMSD | 0.47 | 0.17 | 0.06 |
| | CVRMSD | 0.71 | 0.25 | 0.08 |
| Classification | LC | 51.12 \| 22.92 | 88.61 \| 56.74 | 90.50 \| 59.01 |
| | NC | 50.70 \| 22.16 | 89.94 \| 60.20 | 91.56 \| 61.49 |
| | EC | 92.76 \| 40.54 | 92.76 \| 40.54 | 92.76 \| 40.54 |

Table 35: MaWa and WaCo crop-invariant encoders on supervised ResNet50 features.

| Structure | | STL10 + ResNet50 + Crop | | |
|---|---|---|---|---|
| Preservation | | MaWa ($d = 2048$) | WaCo ($d = 2048$) | WaCo ($d = 512$) |
| Similarity | $L_1$ \| $L_2$ | 0.00 \| 0.77 | 0.50 \| 1.67 | 0.61 \| 1.32 |
| Tests | $m$ \| $b$ | 0.50 \| -12.80 | 1.34 \| -17.20 | 1.09 \| -5.36 |
| | $R^2$ | 0.24 | 0.87 | 0.92 |
| | RMSD | 48.22 | 9.29 | 3.92 |
| | NRMSD | 0.47 | 0.09 | 0.04 |
| | CVRMSD | 0.71 | 0.14 | 0.06 |
| Classification | LC | 50.90 \| 47.93 | 90.77 \| 89.29 | 91.41 \| 90.11 |
| | NC | 50.85 \| 47.69 | 91.71 \| 90.42 | 92.47 \| 91.03 |
| | EC | 92.76 \| 90.73 | 92.76 \| 90.73 | 92.76 \| 90.73 |

Table 36: MaWa and WaCo rotation-invariant encoders on TinyImageNet DINO features.

| Structure | | TinyImageNet + DINO + Rotation | |
|---|---|---|---|
| Preservation | | MaWa ($d = 384$) | WaCo ($d = 384$) |
| Similarity | $L_1$ \| $L_2$ | 0.64 \| 1.18 | 0.46 \| 1.94 |
| Tests | $m$ \| $b$ | 0.99 \| -0.19 | 0.87 \| 0.92 |
| | $R^2$ | 0.84 | 0.29 |
| | RMSD | 2.59 | 15.07 |
| | NRMSD | 0.03 | 0.17 |
| | CVRMSD | 0.02 | 0.14 |
| Classification | LC | 68.71 \| 52.31 | 63.37 \| 43.13 |
| | NC | 70.19 \| 51.24 | 61.26 \| 42.20 |
| | EC | 69.93 \| 27.23 | 69.93 \| 27.23 |

Table 37: MaWa and WaCo affine-invariant encoders on TinyImageNet DINO features.

| Structure | | TinyImageNet + DINO + Affine | |
|---|---|---|---|
| Preservation | | MaWa ($d = 384$) | WaCo ($d = 384$) |
| Similarity | $L_1$ \| $L_2$ | 0.75 \| 1.15 | 0.43 \| 3.22 |
| Tests | $m$ \| $b$ | 0.98 \| 1.83 | 0.92 \| -3.45 |
| | $R^2$ | 0.90 | 0.34 |
| | RMSD | 1.95 | 13.51 |
| | NRMSD | 0.02 | 0.15 |
| | CVRMSD | 0.02 | 0.13 |
| Classification | LC | 68.54 \| 61.83 | 66.89 \| 55.10 |
| | NC | 70.75 \| 60.51 | 65.62 \| 54.55 |
| | EC | 69.93 \| 47.70 | 69.93 \| 47.70 |

Table 38: MaWa and WaCo noise-invariant encoders on TinyImageNet DINO features.

| Structure | | TinyImageNet + DINO + Noise | |
| --- | --- | --- | --- |
| Preservation | | MaWa ($d = 384$) | WaCo ($d = 384$) |
| Similarity | $L_1 \mid L_2$ | 0.86 \| 1.19 | 0.52 \| 1.70 |
| Tests | $m \mid b$ | 0.98 \| 1.77 | 0.87 \| 2.74 |
| | $R^2$ | 0.95 | 0.30 |
| | RMSD | 1.27 | 13.04 |
| | NRMSD | 0.01 | 0.14 |
| | CVRMSD | 0.01 | 0.12 |
| Classification | LC | 68.52 \| 38.38 | 63.62 \| 27.93 |
| | NC | 70.44 \| 37.30 | 62.15 \| 28.67 |
| | EC | 69.93 \| 16.59 | 69.93 \| 16.59 |

Table 39: MaWa and WaCo crop-invariant encoders on TinyImageNet DINO features.

| Structure | | TinyImageNet + DINO + Crop | |
| --- | --- | --- | --- |
| Preservation | | MaWa ($d = 384$) | WaCo ($d = 384$) |
| Similarity | $L_1 \mid L_2$ | 0.82 \| 1.03 | 0.45 \| 1.66 |
| Tests | $m \mid b$ | 0.96 \| 4.01 | 1.02 \| -14.40 |
| | $R^2$ | 0.94 | 0.42 |
| | RMSD | 1.43 | 13.62 |
| | NRMSD | 0.02 | 0.15 |
| | CVRMSD | 0.01 | 0.13 |
| Classification | LC | 68.42 \| 73.43 | 67.20 \| 70.37 |
| | NC | 70.49 \| 72.36 | 66.93 \| 69.30 |
| | EC | 69.93 \| 70.26 | 69.93 \| 70.26 |

Table 40: MaWa and WaCo rotation-invariant encoders on TinyImageNet subset (50 classes) DINO features.

| Structure | | TinyImageNetSubset + DINO + Rotation | |
| --- | --- | --- | --- |
| Preservation | | MaWa ($d = 384$) | WaCo ($d = 384$) |
| Similarity | $L_1 \mid L_2$ | 0.71 \| 1.13 | 0.43 \| 1.40 |
| Tests | $m \mid b$ | 0.99 \| 0.05 | 1.27 \| -32.94 |
| | $R^2$ | 0.84 | 0.41 |
| | RMSD | 2.97 | 10.25 |
| | NRMSD | 0.03 | 0.12 |
| | CVRMSD | 0.03 | 0.10 |
| Classification | LC | 91.52 \| 83.59 | 89.12 \| 77.39 |
| | NC | 93.24 \| 84.70 | 90.60 \| 80.50 |
| | EC | 92.60 \| 53.17 | 92.60 \| 53.17 |

Table 41: MaWa and WaCo affine-invariant encoders on TinyImageNet subset (50 classes) DINO features.

| Structure Preservation | | TinyImageNetSubset + DINO + Affine | |
| --- | --- | --- | --- |
| | | MaWa ($d = 384$) | WaCo ($d = 384$) |
| Similarity Tests | $L_1 \mid L_2$ | 0.80 \| 1.10 | 0.51 \| 1.41 |
| | $m \mid b$ | 0.97 \| 2.32 | 1.19 \| -24.64 |
| | $R^2$ | 0.90 | 0.43 |
| | RMSD | 2.40 | 9.12 |
| | NRMSD | 0.03 | 0.10 |
| | CVRMSD | 0.02 | 0.09 |
| Classification | LC | 91.88 \| 88.18 | 89.96 \| 84.46 |
| | NC | 92.64 \| 88.22 | 91.20 \| 85.96 |
| | EC | 92.60 \| 76.97 | 92.60 \| 76.97 |

Table 42: MaWa and WaCo noise-invariant encoders on TinyImageNet subset (50 classes) DINO features.

| Structure Preservation | | TinyImageNetSubset + DINO + Noise | |
| --- | --- | --- | --- |
| | | MaWa ($d = 384$) | WaCo ($d = 384$) |
| Similarity Tests | $L_1 \mid L_2$ | 0.90 \| 1.20 | 0.46 \| 1.48 |
| | $m \mid b$ | 0.99 \| 1.24 | 1.28 \| -32.81 |
| | $R^2$ | 0.95 | 0.40 |
| | RMSD | 1.38 | 10.15 |
| | NRMSD | 0.02 | 0.12 |
| | CVRMSD | 0.01 | 0.09 |
| Classification | LC | 91.44 \| 72.98 | 89.04 \| 64.38 |
| | NC | 93.16 \| 73.62 | 90.64 \| 68.26 |
| | EC | 92.60 \| 44.25 | 92.60 \| 44.25 |

Table 43: MaWa and WaCo crop-invariant encoders on TinyImageNet subset (50 classes) DINO features.

| Structure Preservation | | TinyImageNetSubset + DINO + Crop | |
| --- | --- | --- | --- |
| | | MaWa ($d = 384$) | WaCo ($d = 384$) |
| Similarity Tests | $L_1 \mid L_2$ | 0.84 \| 1.03 | 0.52 \| 1.69 |
| | $m \mid b$ | 0.96 \| 3.63 | 1.31 \| -36.27 |
| | $R^2$ | 0.93 | 0.41 |
| | RMSD | 2.01 | 10.13 |
| | NRMSD | 0.02 | 0.12 |
| | CVRMSD | 0.02 | 0.09 |
| Classification | LC | 91.56 \| 93.44 | 89.64 \| 91.47 |
| | NC | 93.20 \| 94.47 | 91.36 \| 92.36 |
| | EC | 92.60 \| 92.72 | 92.60 \| 92.72 |

Table 44: Collision rates for R-DINO features on STL10 before (raw) and after rigid alignment.

| Augmentation | CR (%, raw) | CR (%, aligned) |
|---|---|---|
| Rotation | 52.60 | 1.90 |
| Affine | 22.00 | 1.90 |
| Noise | 69.90 | 10.60 |
| Crop | 0.00 | 0.00 |

Table 45: Collision rates for CLIP features on STL10 before (raw) and after rigid alignment.

| Augmentation | CR (%, raw) | CR (%, aligned) |
|---|---|---|
| Rotation | 10.50 | 1.50 |
| Affine | 1.00 | 0.20 |
| Noise | 54.60 | 23.40 |
| Crop | 0.00 | 0.00 |

Table 46: Collision rates for DINO features on TinyImageNet before (raw) and after rigid alignment.

| Augmentation | CR (%, raw) | CR (%, aligned) |
|---|---|---|
| Rotation | 37.92 | 7.90 |
| Affine | 6.53 | 1.55 |
| Noise | 61.91 | 20.39 |
| Crop | 0.05 | 0.04 |

Table 47: Collision rates for DINO features on TinyImageNet subset (50 classes) before (raw) and after rigid alignment.

| Augmentation | CR (%, raw) | CR (%, aligned) |
|---|---|---|
| Rotation | 27.20 | 0.84 |
| Affine | 3.72 | 0.44 |
| Noise | 40.00 | 6.08 |
| Crop | 0.04 | 0.00 |

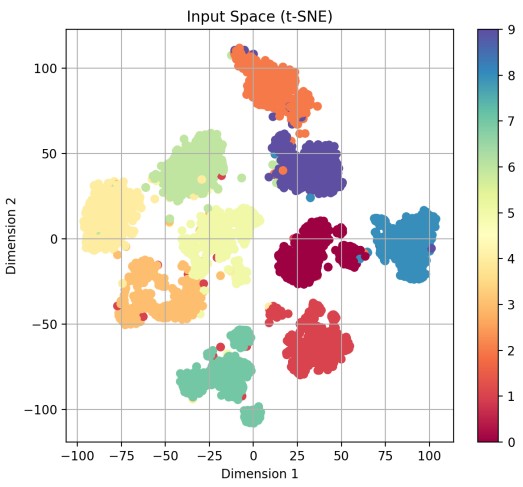

Original DINO Latent Space

MaWa

WaCo

SCLR

HSIC

Figure 5: $t$-SNE visualizations of the original STL10 DINO latent distribution $F_\sharp \mu_X$ compared to $(E_\theta \circ F)_\sharp \mu_X$ for encoders trained for noise invariance with one of the four losses.

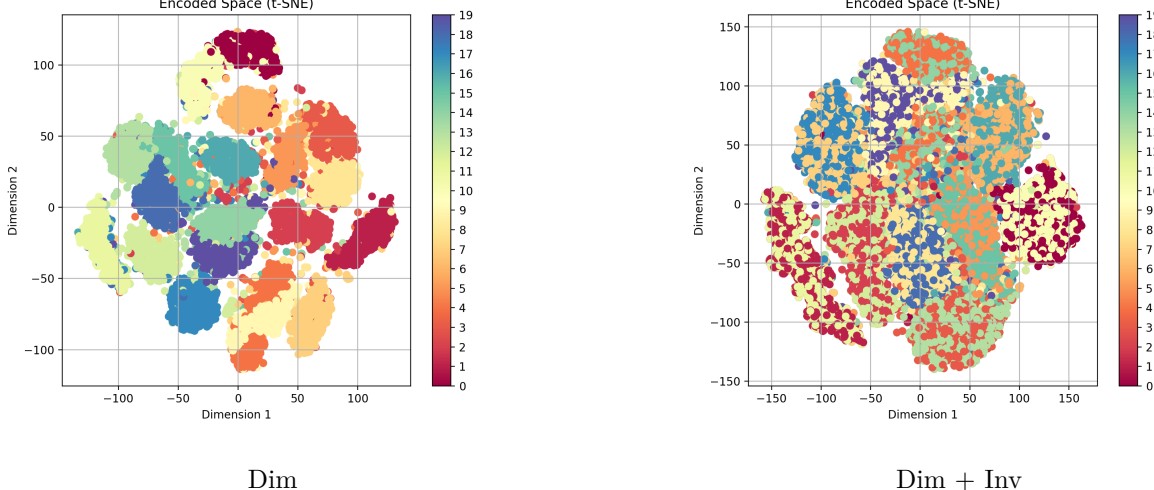

Dim                                                                    Dim + Inv

Figure 6: $t$-SNE visualizations of MNIST plus 90-degree rotated digits for $E_\theta$ with final dimension $d = 64$ trained on the WaCo loss with $T = \{\mathrm{id}_X\}$, which does dimensionality reduction (Dim) only, versus the case of $T = \{\mathrm{id}_X, t\}$, which does Dim plus invariance (Inv) to $t$, where $t$ here is arbitrary rotations.

