# OpenReview forum: "Post-Training Augmentation Invariance"
_TMLR — Accepted by TMLR_

### Review · Reviewer_MEVn · 2025-12-06

**Summary Of Contributions:**

Summary of Contributions
The authors declare the following contributions.

**1. Problem Formalization**
Introduced the post-training augmentation invariance problem, defining a rigorous mathematical framework to add invariance properties to pretrained models without corrupting their original behavior on non-augmented data.
My assessment:
Pros: The conceptualisation of the problem is a useful contribution.
Cons: I am not convinced about the necessity of the theoretical derivation, I find it quite long and hard to graphs the purpose.

**2. Augmented Encoders**
Proposed augmented encoders as a foundational object for augmentation-based learning.
My assessment:
Pros: The conceptualisation of the problem is a useful contribution.
Cons: Again, I find it too long for the content it offers. Moreover, I am somewhat worried about Def 3.1 - see below.

**3. Novel Training Objectives**
Developed two optimal transport-based losses for achieving post-training augmentation invariance MaWa and WaCo.
My assessment:
Pros: The conceptualisation of the problem is a useful contribution. They seem reasonable.
4. Empirical Validation
Demonstrated the effectiveness of MaWa and WaCo through extensive experiments on the STL10 dataset using pretrained models (ResNet50-SwAV and DINOv2).

**My assessment:**
Pros: Shows the benefits of the proposed approach, presents useful comparisons to baselines. Presents insightful analysis of the representation space.
Cons: The experimental part is relatively modest: only one dataset, only one training regime (i.e. number of samples).
**In summary**:
I find the idea of the paper interesting and worth of studying. The presented work is a step in a good direction. However, I am not convinced by the execution. In my view, none of the contribution stands out as the major argument towards accepting the paper.

**Additional Comments:**

An additional question:

To what extent the problem stated is solvable. The authors show some positive results (which is great). However, I'd be curious to see any upper bound. For example, I could imagine that for some augmentation the encoder breaks completely, as it sees OO data, and there is no hope of fixing this with an adapter.

**Audience:**

Yes

**Audience Explanation:**

As said the problem is interesting and relevant.

**Claims And Evidence:**

No

**Claims Explanation:**

Summary of Contributions
The authors declare the following contributions.
1. Problem Formalization
Introduced the post-training augmentation invariance problem, defining a rigorous mathematical framework to add invariance properties to pretrained models without corrupting their original behavior on non-augmented data.
My assessment:
Pros: The conceptualisation of the problem is a useful contribution.
Cons: I am not convinced about the necessity of the theoretical derivation, I find it quite long and hard to graphs the purpose.

2. Augmented Encoders
Proposed augmented encoders as a foundational object for augmentation-based learning.
My assessment:
Pros: The conceptualisation of the problem is a useful contribution.
Cons: Again, I find it too long for the content it offers. Moreover, I am somewhat worried about Def 3.1 - see below.

3. Novel Training Objectives
Developed two optimal transport-based losses for achieving post-training augmentation invariance MaWa and WaCo.
My assessment:
Pros: The conceptualisation of the problem is a useful contribution. They seem reasonable.
4. Empirical Validation
Demonstrated the effectiveness of MaWa and WaCo through extensive experiments on the STL10 dataset using pretrained models (ResNet50-SwAV and DINOv2).
My assessment:
Pros: Shows the benefits of the proposed approach, presents useful comparisons to baselines. Presents insightful analysis of the representation space.
Cons: The experimental part is relatively modest: only one dataset, only one training regime (i.e. number of samples).
**In summary**:
I find the idea of the paper interesting and worth of studying. The presented work is a step in a good direction. However, I am not convinced by the execution. In my view, none of the contribution stands out as the major argument towards accepting the paper.
Are the claims made in the submission supported by accurate, convincing and clear evidence?
No
Contribution 1. A substantial part of the paper is devoted to theory (~6 pages), going quite deeply. However, I fail to see the reason for this, what are insights relevant to the problem that it gives. For example, I do not see why, in the context of the paper, it is important that this or that space is Polish. I’m not strictly against mentioning this, but I do not see any benefit of put this in the main text. Similarly, the loss presented in (43) is quite natural and could be defined without all theory.
Contribution 2. I have concerns about the Def 3.1. It defines the notion of ‘unique augmentations’, however, in practice this notion might be to weak as it operates on full equality.
Two fail cases, I’d imagine:
assume that A are rotations. For a given x, the unique set excludes t=0 (no rotation), but does not exclude rotations arbitrary close to 0. For t = \espilon (a very small number) the representations would probably be very close and the problem mentioned in this section persists.

assume that there are two pictures that are very similar but not equal pixel-wise. They would not generate any exclusions but again might result in the representation space clash.

A question as well, is how the uniqueness is implemented in the experimental part?
Contribution 3. The definition of the augmented autoencoder is valid and useful but quite obvious. Again, I do not see much value from the theoretical derivation.
Contribution 4. The experimental part is, unfortunately, I am not an expert in the area, so I cannot asses this deeply. The choice of the baselines seems valid. I like the in-depth analysis of the representations.
On the negative side:
I encourage the authors to develop a better presentation of Table 1. It is the main results and it is really hard to parse for me. Perhaps adding means over the augmentations, or deltas with respect to the non-augmented case.

I’d love to see a any analysis with respect to the budget (flops/samples). Does the effects stay, or do all methods converge to the same things? etc.

More datasets.

The paper does not contain the standard statistical analysis e.g. confidence intervals or anything that ensures that the results are statistically valid.

**Requested Changes:**

See above.

---

> ### Author Response · Authors · 2026-02-10
> **Proposed Revisions**
>
> Thank you for your thoughtful review and helpful feedback.
>
> Our proposed revisions to the paper are as follows.
>
> $\textbf{Main Experiments}$
>
> We are currently in the process of training additional models and expanding the number of datasets for the experiments section. For our main STL10 tests, we will include comparisons for all of the following models:
>
>         --DINO transformer (already included)
>
>         --SwAV ResNet (already included)
>
>         --DINO ResNet (training mostly done; improvement over baseline still notable, but not as good as the transformer model)
>
>         --CLIP transformer (very strong results for the MaWa loss; Wasserstein correlation still training)
>
>         --Supervised ResNet with final classification head removed (in progress)
>
> We also found that we were too conservative in our initial choice of adapter network. We are still using a single-hidden-layer MLP, but we found that increasing the width of the hidden layer had a positive effect. E.g., our original DINO rotation tests improved from 90 to 94% accuracy with network dimensions (384, 4096, 384), as opposed to the original tests, which kept the hidden layer at the same dimension as the latent space. (We can provide a more complete ablation study on how width and depth affects performance.)
>
> $\textbf{Line Tests}$
>
> We conducted additional synthetic tests to show the crucial role that collisions play in whether or not an invariant can be added to a model. We have a synthetic dataset consisting of images of vertical (V) and horizontal (H) lines. The following tests show what happens when there are collisions, and we also see the crucial role of what's included in the initial distribution.
>
>         --If Aug(V) and Aug(H) are both a single 45 degree counterclockwise rotation, then we can indeed collapse Aug(V) to V and Aug(H) to H while preserving original latent structure.
>
>         --If Aug(V) and Aug(H) include both a single 45 degree counterclockwise rotation (call this Aug_1) and a clockwise 45 degree rotation (Aug_2), then Aug_1(V) = Aug_2(H) and Aug_2(V) = Aug_1(H). In this case, the adapter does indeed leave everything in place, which is what we want, since the only way to obtain invariance in this case would be to collapse the original latent space to a single point.
>
>         --Finally, it's crucial to note that everything is relative to what's included in the original distribution. If we start with V, H, and Aug_1(V), then training for invariance to Aug_1 (45 degree counterclockwise rotation) will only collapse Aug_1(H) to H and will leave V and Aug_1(V) untouched.
>
> $\textbf{Additional Experiments}$
>
> The line tests show what happens when we engineer exact collisions. In practice, approximate overlaps between (augmented) classes in the original latent space are what end up being the fundamental constraint because of the continuity of the encoder. When there is not enough separation in the latent space, a single continuous encoder may be unable to satisfy invariance while keeping the original latent space preserved, and so the final network tends to stay close to the identity. For example, we ran additional tests on TinyImageNet, and when trying to obtain, say, rotation invariance for DINO on all 200 classes simultaneously, the adapter leaves things mostly unchanged. As seen in the t-sne visualization, there is simply too much overlap between the (augmented) classes, which creates competing constraints. On the other hand, we did a test where we selected 50 classes with better separation properties, and then invariance works as desired (strong improvement over baseline, similar to STL10).
>
> We would argue that the above limitation is not so much a limitation of our methods, but rather a fundamental geometric/smoothness constraint on how much a latent space can be reorganized before we inevitably corrupt its original metric structure. If augmented points aug(z_1) and aug(z_2) are close but z_1 and z_2 are far, continuity constrains how easily we can pull aug(z_1) and aug(z_2) apart, and, similarly, if aug(z_1) is close to z_2, then trying to bring aug(z_1) close to z_1 while leaving z_2 fixed is again in tension. In these cases, the network tends to default towards something close to the identity.
>
> $\textbf{Paper Structure}$
>
> Finally, in addition to including more experiments (as outlined above), we propose, in our revision, to move most of the background section into an Appendix. Further, to keep things as accessible as possible, we propose to first present the Markov-Wasserstein loss in its more intuitive form as an anchored MSE loss. The connection with the optimal transport machinery can then be presented in a quick sub-section for anyone who's interested.
>
> Thank you again for your feedback. Please let us know if there is anything else we can do to address your concerns.

---

### Review · Reviewer_gosT · 2025-12-22

**Summary Of Contributions:**

The paper proposes a formal definition of post-training invariance with respect to a set of transformations and a frozen based encoder. Based on this definition the propose a practical way to create invariant representations which seek to simultaneously respect the metric space of the original distribution. To achieve these two opposite objectives they propose two loss functions based on the Wasserstein distance. Finally, they validate their results on several datasets and one pre-trained encoder variant.

**Audience:**

Yes

**Audience Explanation:**

Yes, I believe the findings would be interesting for a wide audience. The need for invariant representation is common one which can't always be addressed before hand since it may not be clear which ones are desired. A method such as the one proposed can help mitigate this problem since practitioners can fit an adaptor network post-hoc.

**Broader Impact Concerns:**

No issues here.

**Claims And Evidence:**

No

**Claims Explanation:**

I like the authors proposed solution to the problem they tackle and believe they made a decent effort at showing that their approach works. My main issue, which I think can be easily addressed, is that they claim that their approach can be used on any pre-trained representation to create (or at the very least approach) an invariant one. However, they only test on one such pre-trained model: DINO. I would like to see other representations being tested, from simple ones like VAEs to more complex ones. I am happy to discuss which representations could be more useful/informative. Otherwise I do think the rest of the analysis is thorough.

**Requested Changes:**

1. The most important for me would be testing with additional pre-trained models. I don't need 20. Just a couple well justified ones.
2. I appreciate the author's effort in including a thorough description of the theory that justifies their approach. However, I don't know if it super helpful for people who are not familiar with these aspects (such as myself) to have it there instead of the appendix. While I can (and did) read it, I haven't been able to develop the necessary intuitions to get much use out of it.

---

> ### Author Response · Authors · 2026-02-10
> **Proposed Revisions**
>
> Thank you for your thoughtful review and helpful feedback.
>
> Our proposed revisions to the paper are as follows.
>
> $\textbf{Main Experiments}$
>
> We are currently in the process of training additional models and expanding the number of datasets for the experiments section. For our main STL10 tests, we will include comparisons for all of the following models:
>
>         --DINO transformer (already included)
>
>         --SwAV ResNet (already included)
>
>         --DINO ResNet (training mostly done; improvement over baseline still notable, but not as good as the transformer model)
>
>         --CLIP transformer (very strong results for the MaWa loss; Wasserstein correlation still training)
>
>         --Supervised ResNet with final classification head removed (in progress)
>
> We also found that we were too conservative in our initial choice of adapter network. We are still using a single-hidden-layer MLP, but we found that increasing the width of the hidden layer had a positive effect. E.g., our original DINO rotation tests improved from 90 to 94% accuracy with network dimensions (384, 4096, 384), as opposed to the original tests, which kept the hidden layer at the same dimension as the latent space. (We can provide a more complete ablation study on how width and depth affects performance.)
>
> $\textbf{Line Tests}$
>
> We conducted additional synthetic tests to show the crucial role that collisions play in whether or not an invariant can be added to a model. We have a synthetic dataset consisting of images of vertical (V) and horizontal (H) lines. The following tests show what happens when there are collisions, and we also see the crucial role of what's included in the initial distribution.
>
>         --If Aug(V) and Aug(H) are both a single 45 degree counterclockwise rotation, then we can indeed collapse Aug(V) to V and Aug(H) to H while preserving original latent structure.
>
>         --If Aug(V) and Aug(H) include both a single 45 degree counterclockwise rotation (call this Aug_1) and a clockwise 45 degree rotation (Aug_2), then Aug_1(V) = Aug_2(H) and Aug_2(V) = Aug_1(H). In this case, the adapter does indeed leave everything in place, which is what we want, since the only way to obtain invariance in this case would be to collapse the original latent space to a single point.
>
>         --Finally, it's crucial to note that everything is relative to what's included in the original distribution. If we start with V, H, and Aug_1(V), then training for invariance to Aug_1 (45 degree counterclockwise rotation) will only collapse Aug_1(H) to H and will leave V and Aug_1(V) untouched.
>
> $\textbf{Additional Experiments}$
>
> The line tests show what happens when we engineer exact collisions. In practice, approximate overlaps between (augmented) classes in the original latent space are what end up being the fundamental constraint because of the continuity of the encoder. When there is not enough separation in the latent space, a single continuous encoder may be unable to satisfy invariance while keeping the original latent space preserved, and so the final network tends to stay close to the identity. For example, we ran additional tests on TinyImageNet, and when trying to obtain, say, rotation invariance for DINO on all 200 classes simultaneously, the adapter leaves things mostly unchanged. As seen in the t-sne visualization, there is simply too much overlap between the (augmented) classes, which creates competing constraints. On the other hand, we did a test where we selected 50 classes with better separation properties, and then invariance works as desired (strong improvement over baseline, similar to STL10).
>
> We would argue that the above limitation is not so much a limitation of our methods, but rather a fundamental geometric/smoothness constraint on how much a latent space can be reorganized before we inevitably corrupt its original metric structure. If augmented points aug(z_1) and aug(z_2) are close but z_1 and z_2 are far, continuity constrains how easily we can pull aug(z_1) and aug(z_2) apart, and, similarly, if aug(z_1) is close to z_2, then trying to bring aug(z_1) close to z_1 while leaving z_2 fixed is again in tension. In these cases, the network tends to default towards something close to the identity.
>
> $\textbf{Paper Structure}$
>
> Finally, in addition to including more experiments (as outlined above), we propose, in our revision, to move most of the background section into an Appendix. Further, to keep things as accessible as possible, we propose to first present the Markov-Wasserstein loss in its more intuitive form as an anchored MSE loss. The connection with the optimal transport machinery can then be presented in a quick sub-section for anyone who's interested.
>
> Thank you again for your feedback. Please let us know if there is anything else we can do to address your concerns.

---

### Review · Reviewer_oc4U · 2026-01-31

**Summary Of Contributions:**

This paper introduces a principled framework for post-training augmentation invariance: enhancing a frozen pretrained network’s robustness to specific input transformations without altering its behavior on the original data distribution. The authors formalize this via (t, µₓ, F, V)-invariance, define augmented encoders as probabilistic models of augmentation-encoding processes, and propose two optimal transport-based losses—Markov-Wasserstein minimization and Wasserstein correlation maximization—to train lightweight MLP adapters appended to the latent space of frozen feature extractors. Empirically, on STL10, their method boosts rotation-invariant accuracy and noise-invariant accuracy. Crucially, they demonstrate that alternative losses fail to achieve comparable invariance and significantly corrupt pretrained representations.

Strengths
1. Clearly distinguishes post-training invariance from standard fine-tuning or representation learning, addressing a practical gap in adapting foundation models.
2. Integrates measure theory and optimal transport to formalize invariance constraints and adapter behavior.
3. Achieves gains using minimal compute (single-layer MLP adapters, no backbone updates), enhancing deployability.

Weaknesses
1. Experiments focus solely on STL10 with two backbone architectures; broader validation across datasets, modalities (e.g., medical/satellite imagery), or augmentation types (e.g., affine transforms) would strengthen claims.
2. Optimal transport losses may scale poorly to high-dimensional latent spaces or large batch sizes; runtime/memory comparisons against baselines are absent.
3. The choice of a one-hidden-layer MLP is empirically justified but lacks theoretical motivation; sensitivity to adapter capacity is unexplored.

**Audience:**

Yes

**Audience Explanation:**

The problem of inference invariance is widely known by researchers and some actual users. It is hard to solve and the  fragile robustness of trained models has attracted many attacks. This paper posts a post training method to solve it.

**Broader Impact Concerns:**

No.

**Claims And Evidence:**

Yes

**Claims Explanation:**

The proposed losses (Markov-Wasserstein minimization, Wasserstein correlation maximization) enable post-training invariance. Clear metrics (accuracy on rotated/noisy images) demonstrate substantial improvements over baselines (SimCLR, HSIC). The 19% absolute gain for rotation invariance and ～18% for noise invariance are statistically meaningful. Direct comparison to alternative losses (SimCLR, HSIC) shows these baselines degrade performance and corrupt latent spaces, validating the uniqueness of the proposed losses.

**Requested Changes:**

The writing of the paper need to be more easy to understand by general researchers.

---

> ### Author Response · Authors · 2026-02-10
> **Proposed Revisions**
>
> Thank you for your thoughtful review and helpful feedback.
>
> Our proposed revisions to the paper are as follows.
>
> $\textbf{Main Experiments}$
>
> We are currently in the process of training additional models and expanding the number of datasets for the experiments section. For our main STL10 tests, we will include comparisons for all of the following models:
>
>         --DINO transformer (already included)
>
>         --SwAV ResNet (already included)
>
>         --DINO ResNet (training mostly done; improvement over baseline still notable, but not as good as the transformer model)
>
>         --CLIP transformer (very strong results for the MaWa loss; Wasserstein correlation still training)
>
>         --Supervised ResNet with final classification head removed (in progress)
>
> We also found that we were too conservative in our initial choice of adapter network. We are still using a single-hidden-layer MLP, but we found that increasing the width of the hidden layer had a positive effect. E.g., our original DINO rotation tests improved from 90 to 94% accuracy with network dimensions (384, 4096, 384), as opposed to the original tests, which kept the hidden layer at the same dimension as the latent space. (We can provide a more complete ablation study on how width and depth affects performance.)
>
> $\textbf{Line Tests}$
>
> We conducted additional synthetic tests to show the crucial role that collisions play in whether or not an invariant can be added to a model. We have a synthetic dataset consisting of images of vertical (V) and horizontal (H) lines. The following tests show what happens when there are collisions, and we also see the crucial role of what's included in the initial distribution.
>
>         --If Aug(V) and Aug(H) are both a single 45 degree counterclockwise rotation, then we can indeed collapse Aug(V) to V and Aug(H) to H while preserving original latent structure.
>
>         --If Aug(V) and Aug(H) include both a single 45 degree counterclockwise rotation (call this Aug_1) and a clockwise 45 degree rotation (Aug_2), then Aug_1(V) = Aug_2(H) and Aug_2(V) = Aug_1(H). In this case, the adapter does indeed leave everything in place, which is what we want, since the only way to obtain invariance in this case would be to collapse the original latent space to a single point.
>
>         --Finally, it's crucial to note that everything is relative to what's included in the original distribution. If we start with V, H, and Aug_1(V), then training for invariance to Aug_1 (45 degree counterclockwise rotation) will only collapse Aug_1(H) to H and will leave V and Aug_1(V) untouched.
>
> $\textbf{Additional Experiments}$
>
> The line tests show what happens when we engineer exact collisions. In practice, approximate overlaps between (augmented) classes in the original latent space are what end up being the fundamental constraint because of the continuity of the encoder. When there is not enough separation in the latent space, a single continuous encoder may be unable to satisfy invariance while keeping the original latent space preserved, and so the final network tends to stay close to the identity. For example, we ran additional tests on TinyImageNet, and when trying to obtain, say, rotation invariance for DINO on all 200 classes simultaneously, the adapter leaves things mostly unchanged. As seen in the t-sne visualization, there is simply too much overlap between the (augmented) classes, which creates competing constraints. On the other hand, we did a test where we selected 50 classes with better separation properties, and then invariance works as desired (strong improvement over baseline, similar to STL10).
>
> We would argue that the above limitation is not so much a limitation of our methods, but rather a fundamental geometric/smoothness constraint on how much a latent space can be reorganized before we inevitably corrupt its original metric structure. If augmented points aug(z_1) and aug(z_2) are close but z_1 and z_2 are far, continuity constrains how easily we can pull aug(z_1) and aug(z_2) apart, and, similarly, if aug(z_1) is close to z_2, then trying to bring aug(z_1) close to z_1 while leaving z_2 fixed is again in tension. In these cases, the network tends to default towards something close to the identity.
>
> $\textbf{Paper Structure}$
>
> Finally, in addition to including more experiments (as outlined above), we propose, in our revision, to move most of the background section into an Appendix. Further, to keep things as accessible as possible, we propose to first present the Markov-Wasserstein loss in its more intuitive form as an anchored MSE loss. The connection with the optimal transport machinery can then be presented in a quick sub-section for anyone who's interested.
>
> Thank you again for your feedback. Please let us know if there is anything else we can do to address your concerns.

---

### Decision · Action_Editor_6xBH · 2026-04-26

**Recommendation:** Accept as is

**Audience:**

Yes

**Audience Explanation:**

The augmentation invariance is a problem that is relevant in practice, and is of interest to the TMLR community.

The reviewers had concerns with the overly theoretical presentation of the paper; reviewers were concerned with whether the theoretical findings are significant and add much to the paper. The authors restructured the paper, so that it is possible to skip some of the more theoretical sections, and some of the material has been moved to appendix, to make the paper more accessible. Additionally, the significance of theoretical results is not a requirement for acceptance to TMLR.

**Claims And Evidence:**

Yes

**Claims Explanation:**

The paper studies the problem of adding augmentation invariance to classifiers in post-training. The authors develop a theoretical framework for post-training invariance via optimal transport, with precise definitions. They ultimately propose an intuitive method, MaWa, which trains an adaptor model on top of a pretrained classifier representation space to add invariance (as well as another method, WaCo).

One of the initial concerns of the reviewers was the limited empirical evaluation. The authors now added results to validate their method across multiple architectures and pretrained models, with consistent results.

The paper would still benefit from more realistic large-scale datasets (currently, only STL-10 is used), but overall the evaluation is adequate.